# A quantitative framework reveals ecological drivers of grassland microbial community assembly in response to warming

Daliang Ning [1,2], Mengting Yuan [1,3], Linwei Wu [1], Ya Zhang[1], Xue Guo [1,2], Xishu Zhou[1,4], Yunfeng Yang [2], Adam P. Arkin[5,6], Mary K. Firestone[3,7] & Jizhong Zhou [1,2,7,8 ✉]

Unraveling the drivers controlling community assembly is a central issue in ecology. Although it is generally accepted that selection, dispersal, diversification and drift are major community assembly processes, defining their relative importance is very challenging. Here, we present a framework to quantitatively infer community assembly mechanisms by phylogenetic bin-based null model analysis (iCAMP). iCAMP shows high accuracy (0.93–0.99), precision (0.80–0.94), sensitivity (0.82–0.94), and specificity (0.95–0.98) on simulated communities, which are 10–160% higher than those from the entire community-based approach. Application of iCAMP to grassland microbial communities in response to experimental warming reveals dominant roles of homogeneous selection (38%) and 'drift' (59%). Interestingly, warming decreases 'drift' over time, and enhances homogeneous selection which is primarily imposed on Bacillales. In addition, homogeneous selection has higher correlations with drought and plant productivity under warming than control. iCAMP provides an effective and robust tool to quantify microbial assembly processes, and should also be useful for plant and animal ecology.

[1] Institute for Environmental Genomics and Department of Microbiology and Plant Biology, University of Oklahoma, Norman, OK 73019, USA. [2] State Key Joint Laboratory of Environment Simulation and Pollution Control, School of Environment, Tsinghua University, 100084 Beijing, China. [3] Department of Environmental Science Policy and Management, University of California, Berkeley, CA 94720, USA. [4] School of Minerals Processing and Bioengineering, Central South University, 410083 Changsha, Hunan, China. [5] Environmental Genomics and Systems Biology, Lawrence Berkeley National Laboratory, Berkeley, CA 94710, USA. [6] Department of Bioengineering, University of California, Berkeley, CA 94720, USA. [7] Earth and Environmental Sciences, Lawrence Berkeley National Laboratory, Berkeley, CA 94704, USA. [8] School of Civil Engineering and Environmental Sciences, University of Oklahoma, Norman, OK 73019, USA. ✉email: jzhou@ou.edu

Microorganisms are the most diverse groups of life presently known, inhabiting almost every imaginable environment on the Earth, and they typically form complex communities whose structure, functions, interactions, and dynamics are critical to ecosystem functioning and service. However, characterizing such complex communities, quantifying the accompanying ecological processes, and dissecting the mechanisms controlling biodiversity and community composition are extremely challenging[1]. With the rapid development of high-throughput metagenomic technologies[1], large experimental data on community structure can be rapidly obtained. However, analyzing such massive data to address fundamental ecological questions, such as community assembly mechanisms is challenging due to various issues associated with detection specificity, sensitivity, quantification, reproducibility, and taxonomic resolution[1].

Understanding community assembly rules is a longstanding issue of ecologists[2,3]. Niche-based theory asserts that deterministic processes, including environmental filtering (e.g., pH, temperature, moisture, and salinity) and various biological interactions (e.g., competition, facilitation, mutualisms, and predation), largely control the patterns of species composition, abundance, and distributions[4,5]. By contrast, neutral theory assumes that all species are ecologically equivalent, and species dynamics are largely controlled by stochastic processes of birth/death, speciation/extinction, and immigration[4,6]. After intensive debates in 2000s[6–8], it is generally accepted that both deterministic and stochastic processes operate simultaneously in the assembly of local communities[8–10], and the key question becomes how to define their relative importance in controlling community diversity, distribution, and succession[3,8–14].

To unify niche and neutral perspectives on governing community structure, Vellend[12,15] proposed a conceptual framework that community diversity and dynamics are controlled by four high-level general ecological processes: selection, dispersal, speciation or diversification, and ecological drift[15,16]. Hereafter, we use the term 'ecological processes' particularly to represent these community assembly processes. Although the framework has recently received a great attention in microbial ecology[3,17–19], translating this conceptual framework into a quantitative operational model is even more challenging[16–18,20]. Due to the lack of quantitative approaches, most analyses with respect to the relative importance of the four processes across different types of natural communities are qualitative and subjective, and replete with great uncertainty[16]. As an exploratory effort, a null modeling-based operational approach was developed to obtain quantitative information on community assembly processes from the statistical perspective[19,20], which is abbreviated as QPEN (Quantifying assembly Processes based on Entire-community Null model analysis) hereafter. QPEN uses phylogenetic metrics to infer selection since phylogenetic distance could reflect niche difference (so-called phylogenetic signal) within some threshold[19,21].

This statistical approach represents a significant advance in microbial ecology that enables microbial ecologists to obtain quantitative information on community assembly processes[3]. It has provided valuable insights into the importance of various ecological processes in microbial ecology[19,20,22–26] and plant ecology[27]. However, a major limitation is that various ecological processes are estimated based on the pairwise turnovers of the whole communities[19,20]. This may not be appropriate because it is well known that the actions of various ecological processes (e.g. natural selection) are typically on the finer biological organization levels, such as genotypes and populations rather than whole communities[10,17,18,28,29]. Within a single microbial community, certain populations are under strong selection, whereas others could be under strong drift. This type of difference cannot be discerned using whole community level metrics. Also, various

groups of organisms differ greatly in their responses to environmental changes. Similarly, the dispersal ability, diversification rates, and susceptibility to drift are substantially different among various microbial groups. Thus, it would be meaningful to consider selection and other ecological processes at the level of individual taxa/lineages rather than the entire community[3,17]. To this end, we developed a general framework to quantitatively infer Community Assembly Mechanisms by Phylogenetic-bin-based null model analysis, abbreviated as iCAMP, based on the turnovers of individual bins across communities (samples). We apply this approach to investigate whether and how experimental warming affects various ecological processes in the assembly of grassland soil microbial communities. Our results indicate that iCAMP provides a robust, reliable tool for quantifying the relative importance of ecological processes in controlling microbial community diversity and succession.

## Results

**Overview of iCAMP.** To quantify various ecological processes, the observed taxa are first divided into different groups ('bins') based on their phylogenetic relationships (Fig. 1, Supplementary Fig. 1a). Then, the process governing each bin is identified based on null model analysis of the phylogenetic diversity using beta Net Relatedness Index (βNRI), and taxonomic β-diversities using modified Raup–Crick metric (RC) (Fig. 1, Supplementary Fig. 1b). For each bin, the fraction of pairwise comparisons with βNRI < −1.96 is considered as the percentages of homogeneous

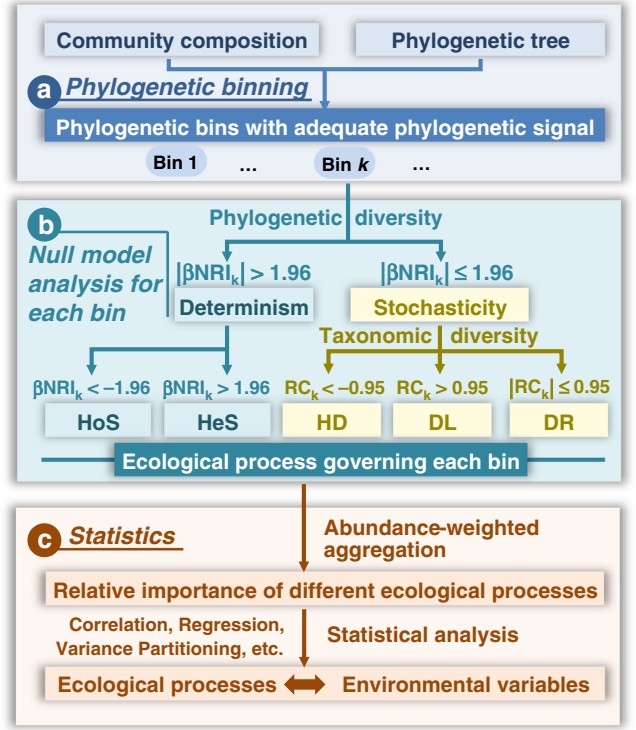

**Fig. 1 Overview of iCAMP.** iCAMP includes several key steps: **a** phylogenetic binning; **b** bin-based null model simulations with phylogenetic diversity for partitioning selection, and taxonomic diversity for partitioning dispersal and drift; and **c** statistical analysis for assessing relative importance of different ecological processes and linking the processes with different environmental factors. βNRI beta net relatedness index, RC modified Raup–Crick metric; Here, 'ecological processes' particularly mean community assembly processes, including homogeneous selection (HoS), heterogeneous selection (HeS), homogenizing dispersal (HD), dispersal limitation (DL), and 'drift' (DR). See the main text for a detailed explanation.

selection, whereas those with βNRI > +1.96 as the percentages of heterogeneous selection based on the threshold applied previously[20,30]. Next, taxonomic diversity metric RC is used to partition the remaining pairwise comparisons with |βNRI| ≤ 1.96. The fraction of pairwise comparisons with RC < −0.95 is treated as the percentages of homogenizing dispersal, while those with RC > + 0.95 as dispersal limitation[19,20]. The remains with |βNRI| ≤ 1.96 and |RC| ≤ 0.95 represent the percentages of drift, diversification, weak selection, and/or weak dispersal[3], hereafter, simply designated as 'drift'[19] for convenience. The above analysis is repeated for every bin. Subsequently, the fractions of individual processes across all bins are weighted by the relative abundance of each bin, and summarized to estimate the relative importance of individual processes at the whole community level (Fig. 1, Supplementary Fig. 1e). Furthermore, various statistical analyses, e.g. Mantel test, multiple regression on distance matrix (MRM), variation partitioning, etc., are used to reveal the linkages of individual processes to different environmental factors for obtaining detailed insights into community assembly mechanisms (Fig. 1). Besides βNRI and RC, iCAMP also incorporated direct test based on null model distribution, which provided highly similar results as βNRI and RC in this study (Supplementary Fig. 17), and should be a preferred choice when the null model simulated values do not follow normal distribution[31] (see Supplementary Note 1 for details).

**Simulated communities**. Due to lack of a gold-standard experiment to establish the true community assembly processes, using simulated communities is the predominant approach for assessing performances of various computational methods, as shown in many other computational studies[32,33]. Thus, we used a simulation model to generate communities with pre-defined relative importance of each ecological process (Supplementary Fig. 2). In this simulation model, four plots (LA, LB, HA, HB) in two islands (A, B) under two types of environments (L, H) are considered, with six local communities sampled from each plot (Supplementary Fig. 2a). Each local community consists of different types of species controlled by drift, selection, or dispersal (Supplementary Fig. 2c–e).

Three scenarios were simulated with three different levels of phylogenetic signal in the regional species pool: low (Blomberg's $K = 0.15$), medium ($K = 0.9$), and high ($K = 5.5$). Each scenario has 15 simulated situations, where the expected ('true') relative importance of each process (selection, dispersal, drift) was set from 0% to 100% (with an interval of 25%, Supplementary Fig. 2b and Supplementary Table 1). Then, the relative importance of different ecological processes was assessed by iCAMP. The performance was evaluated with six quantitative and qualitative indexes: quantitative accuracy and precision, and qualitative accuracy, precision, sensitivity, and specificity (detailed in the Methods section, Eqs. (16)–(21)). The quantitative performance indexes are based on the difference between the expected and estimated relative importance of each process, while the qualitative performance indexes are calculated from the true or false identification of dominant process. According to the performances with simulated communities, we optimized the binning algorithms (Supplementary Figs. 3 and 4), metrics (Supplementary Fig. 5), and null model algorithms (Supplementary Fig. 6), and explored the impact of randomization times and resampling taxa on the performance (Supplementary Figs. 7 and 8; detailed in Supplementary Note 1).

**Comparison between iCAMP and other approaches**. After appropriate parameter settings were determined, iCAMP and several previously reported approaches were compared for their performances with the simulated communities (Fig. 2, Supplementary Figs. 9–12). First, the ecological stochasticity was quantified by five approaches, including abundance-weighted neutral taxa percentage (NP)[3] based on neutral-theory model, normalized stochasticity ratios (NST)[33] based on taxonomic (tNST) or phylogenetic metrics (pNST), and the relative importance of stochastic processes (homogenizing dispersal, dispersal limitation, and drift) based on QPEN or iCAMP. Under high-phylogenetic-signal and medium-phylogenetic-signal scenarios, iCAMP consistently showed the highest quantitative accuracy (0.978–0.997) and precision (0.903–0.930), while pNST exhibited similar accuracy (0.924–0.954) but lower precision (0.658–0.722, $p < 0.001$, Fig. 2a–c). Under low-phylogenetic-signal scenario, iCAMP continued to show the highest precision (0.807) and the second-high accuracy (0.770), while pNST showed similar precision (0.723) and the highest accuracy (0.947). By contrast, tNST, NP, and QPEN showed lower precision (<0.57, down to −0.75, $p < 0.0001$) than iCAMP in simulated scenarios (Fig. 2a–c).

Only QPEN and iCAMP can quantify relative importance of different ecological processes, so we compared their performances. On average, iCAMP had higher accuracy (0.93–0.99 against 0.81–0.97, 9.9% higher), precision (0.82–0.94 against 0.33–0.52, 120.2% higher), sensitivity (0.83–0.94 against 0.54–0.58, 61.1% higher), and specificity (0.96–0.98 against 0.87–0.88, 10.6% higher) than QPEN (Fig. 2i, Supplementary Fig. 9k, l). The setting of phylogenetic signal also had significant impacts on iCAMP performance. When the phylogenetic signal increased from low/medium to high (Fig. 2i, Supplementary Fig. 9k, l), the accuracy and specificity of iCAMP remained high (>0.92) without significant changes ($p > 0.20$), but the precision and sensitivity of iCAMP increased from 0.80–0.82 to 0.90–0.94. By contrast, the overall performance of QPEN was improved by higher phylogenetic signal. In addition, iCAMP demonstrated good robustness to the uncertainty in bin determination (Supplementary Figs. 10 and 11, see Supplementary Note 2).

The performance varied considerably among different ecological processes (Fig. 2d–h, Supplementary Fig. 12). In the simulated communities under medium and high phylogenetic signals, all performance indices were higher than 0.78 for iCAMP (Supplementary Fig. 12), indicating considerable improvement from QPEN, particularly in estimating homogeneous and heterogeneous selections. However, with low phylogenetic signal, iCAMP had low sensitivity (down to 0.17) for homogeneous selection (Supplementary Fig. 12), albeit still higher ($p < 0.05$) than QPEN (sensitivity < 0.04). These results confirmed that low phylogenetic signal of niche preference can limit the capability of phylogenetic metrics to infer selection, which can be partly but not completely overcome by iCAMP. Nevertheless, the quantitative performance of iCAMP remained relatively high for all processes under all scenarios, with quantitative accuracy and precision 0.71–1.00 (averagely 129% higher than QPEN), indicating that iCAMP can substantially improve the quantitative estimation of community assembly processes.

**Effects of warming on grassland bacterial assembly**. To determine the effectiveness of iCAMP in real-world studies, iCAMP was applied to an empirical data of soil bacterial communities in a grassland under experimental warming[34], with focus on within-treatment spatial turnovers. Based on iCAMP analysis, homogeneous selection and drift were more important than other processes in bacterial community assembly, with average relative importance of 37.0–38.5% and 58.3–59.9% (Fig. 3a, b), respectively. Warming significantly altered the relative importance of different processes ($p < 0.01$, permutational ANOVA). Since other processes had quite low estimated relative importance (<3.4%), we primarily

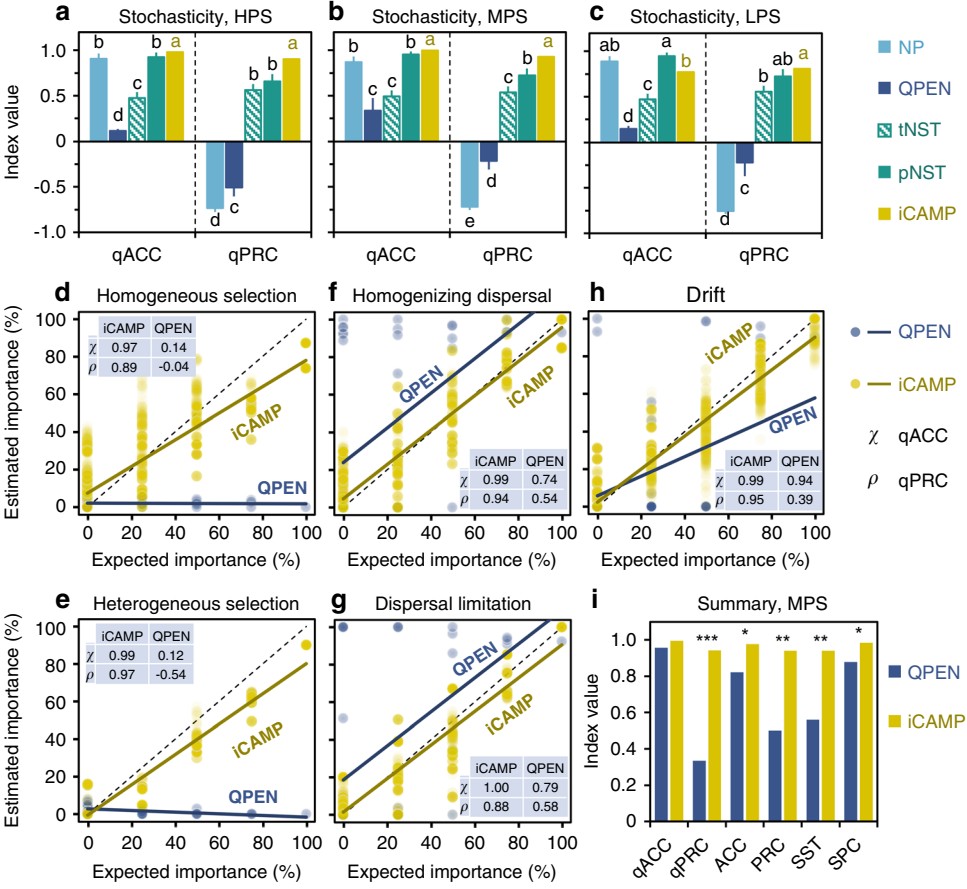

**Fig. 2 Performances of iCAMP and other approaches with simulated communities. a–c** The performances of different approaches in quantifying ecological stochasticity under high-phylogenetic-signal (HPS), medium-phylogenetic-signal (MPS), and low-phylogenetic-signal (LPS) scenarios, which were assessed by quantitative accuracy (qACC) and precision (qPRC). Data are presented as mean values ± SD. Error bars indicate standard deviations based on $n = 1000$ times bootstrapping from 15 independent situations. **d–h** The performances of iCAMP ($n = 4140$ comparisons = 276 comparisons among 24 biologically independent samples in each of 15 situations) and QPEN ($n = 60$ groups = 4 groups of comparisons among 24 biologically independent samples in each of 15 situations) were also evaluated based on the consistency between the estimated and expected relative importance of different individual ecological processes under MPS scenario. **i** Overall performance of iCAMP ($n = 4140$) and QPEN ($n = 60$) under MPS scenario assessed with six performance indexes: qACC ($\chi$), qPRC ($\rho$), qualitative accuracy (ACC), qualitative precision (PRC), sensitivity (SST), and specificity (SPC). QPEN quantifying assembly processes based on entire-community null model analysis; NP abundance-weighted neutral taxa percentage; tNST and pNST taxonomic and phylogenetic normalized stochasticity ratio. One-side significance based on bootstrapping test ($n = 1000$) was indicated by **a–c** different letters ($p < 0.05$, see Source Data file for exact $p$ values), or **i** ***$p < 0.001$; **$p < 0.01$; *$p < 0.05$ ($p = 0.130, 0.000, 0.023, 0.007, 0.008, 0.031$ for qACC, qPRC, ACC, PRC, SST, SPC, respectively). Source data are provided as a Source Data file.

focused on the effects of warming on homogeneous selection and drift in subsequent analyses. Overall, warming decreased the relative importance of drift and increased homogeneous selection. Significant year-to-year variations were observed (Fig. 3c, d). In the first year, the communities under warming showed significantly higher ratio of drift (Cohen's $d = 2.9$, $p = 0.001$), but lower ratio of homogeneous selection (Cohen's $d = −2.7$, $p < 0.001$) than those under control, suggesting the bacterial community assembly was even more stochastic under warming than control in the beginning. In the second year, the difference between warming and control became insignificant. In the third to fifth years, the communities under warming had significantly higher ratio of homogeneous selection (Cohen's $d = 0.6–1.7$) and lower ratio of drift (Cohen's $d = −0.8$ to $−1.3$), suggesting that the selection pressure imposed by warming on the soil bacteria gradually increased with time.

QPEN was also applied to quantify the ecological processes. The results from QPEN indicated that homogeneous selection predominated (>73%) bacterial assembly with higher relative importance under warming (83.3%) than control (73.3%, Supplementary

Fig. 13a, b) although not significant ($p = 0.174$). QPEN suggested 0.0% of drift under warming, 0.0% of heterogeneous selection, and homogenizing dispersal across all years, and 100% of homogeneous selection in some years (Supplementary Fig. 13c, d). This appears not reasonable, considering important roles of stochastic processes have been widely reported across various ecosystems[10,11,13,14,35], including desert[28] and acidic soils[25].

**Stochastic vs. deterministic bacterial assembly.** Based on the principle of the null models employed by iCAMP and QPEN, the fractions of dispersal limitation, homogenizing dispersal, and drift are largely considered stochastic[3]. Thus the sum of their estimated relative importance can be used to estimate stochasticity of community assembly. Based on iCAMP results, the relative importance of stochastic processes was 62.6% under control and 61.3% under warming (Supplementary Fig. 14). By contrast, QPEN estimated the relative importance of stochastic processes was 26.7% under control and 16.7% under warming, which were much lower than those estimated by other approaches (Supplementary Fig. 14). For instance, variation partitioning analysis

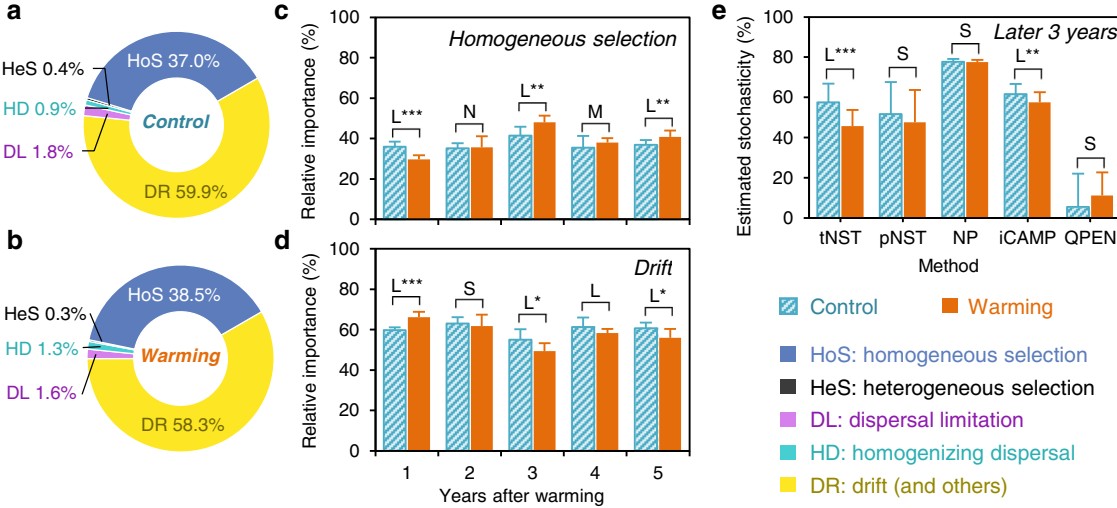

**Fig. 3 Relative importance of different ecological processes in response to warming. a** Under control. **b** Under warming. **c** Changes of homogeneous selection under warming (orange bar) and control (aqua bar). **d** Changes of drift. **a–d** were estimated by iCAMP. **e** Stochasticity estimated by different methods in the later 3 years. One-side significance based on bootstrapping test was expressed as ***$p < 0.01$; **$p < 0.05$; *$p < 0.1$. $p = 0.001$, 0.818, 0.014, 0.346, 0.035 in Year 1–5 for homogeneous selection; $p = 0.001$, 0.657, 0.066, 0.207, 0.058 in Year 1–5 for drift; $p = 0.000$, 0.542, 0.500, 0.014, 0.567 for tNST, pNST, NP, iCAMP, and QPEN, respectively. L, M, S, and N represented large ($|d| > 0.8$), medium ($0.5 < |d| \leq 0.8$), small ($0.2 < |d| \leq 0.5$), and negligible ($|d| \leq 0.2$) effect sizes of warming, based on Cohen's $d$ (the mean difference between warming and control divided by pooled standard deviation). Data are presented as mean values ± SD. Error bars represented standard deviations; **c** and **d** $n = 6$ comparisons among four biologically independent samples at each time point; **e** $n = 18$ comparisons = 6 comparisons in each of the 3 years. Source data are provided as a Source Data file.

(VPA) revealed that substantial portions of the community variations (68.4%) could not be explained by all measured environmental variables[34]. The tNST and pNST were on average 48.8% under warming and 52.3% under control, and NP ranged from 74% to 79% in different years for both warming and control (Supplementary Fig. 14). It appears that VPA, NST, and NP showed more consistent results with iCAMP than QPEN.

All approaches did not reveal significant ($p > 0.10$) differences of the 5-year mean stochasticity between warming and control, except tNST with medium effect size ($p < 0.05$, Supplementary Fig. 14). But in the third to fifth year (Fig. 3e), both tNST and iCAMP revealed that warming had significant ($p < 0.05$) decrease in stochasticity, and there was slight decrease in stochasticity with pNST and NP under warming (small effect size) though it was insignificant ($p > 0.10$). On the contrary, QPEN showed a slight but insignificant increase in stochasticity. Collectively, consistent with our previous analysis[34], various approaches supported that stochastic processes could play more important roles in grassland soil bacterial assembly and that warming decreased the stochasticity after 3 years.

**Assembly mechanisms across different phylogenetic groups.** In contrast to QPEN and other approaches, iCAMP can provide information on the relative importance of different ecological processes in individual lineages (bins). For this purpose, the observed 18,123 OTUs were divided into 658 phylogenetic bins, each of which was then analyzed separately as outlined in Fig. 1. Our results revealed that homogeneous selection dominated 59 bins (9% of bin numbers and 33% of relative abundance, Fig. 4a). Two of the major bins were Bacillales (Bin 1, 26.7% in total abundance of bins controlled by homogeneous selection) in Firmicutes and Spartobacteria (Bin 2, 18.8%) in Verrucomicrobia (Fig. 4b, Supplementary Fig. 15a). By contrast, drift dominated 598 bins (91% of bin numbers and 67% of relative abundance, Fig. 4a), which mainly belonged to Class Alphaproteobacteria (22.2% in total abundance of drift-controlled bins) and Phylum Actinobacteria (23.5%, Supplementary Fig. 15a).

To understand how different lineages respond to warming, we further determined the bacterial groups contributing to the warming-induced changes of homogeneous selection and drift in the third to fifth years (Fig. 4c, d). Our results revealed that Firmicutes contributed 58.2% of the warming-induced increases in homogeneous selection (Supplementary Fig. 15b). The most abundant Firmicutes bin (Bin 1, Bacillales, average 74.8% in Firmicutes) was governed by homogeneous selection (Supplementary Fig. 15c). After Year 3, which had severe drought, Firmicutes were significantly more abundant under warming than control (Supplementary Fig. 15c). 1 by contrast, the decrease of drift under warming was due to similar negative responses of many bins in five phyla (Proteobacteria, Verrucomicrobia, Bacteroidetes, Planctomycetes, and Acidobacteria, Fig. 4c, Supplementary Fig. 15b). For instance, Bin 4 of Rhizobiales in Alphaproteobacteria had lower relative abundance and reduced relative importance of drift under warming, especially in later 3 years (Supplementary Fig. 15d). From Year 3 to Year 5, 27 bins showed significantly higher relative abundances under warming than control (Wilcoxon $p < 0.05$, Cohen's $d > 0.5$). However, 81% of them (22 bins) were mainly governed by drift, while two of them switched from governed by drift in controls to homogeneous selection under warming. These results demonstrated complex assembly mechanisms of different bins in response to warming, and possible misinterpretation could occur without bin-level mechanistic information.

**Environmental factors influencing ecological processes.** The influences of environmental factors on various ecological processes were further determined with Mantel test and MRM. Since the relative importance of other processes (heterogeneous selection, homogenizing dispersal, and dispersal limitation) was small, we primarily focus on homogeneous selection and drift. Overall, these two processes were significantly linked to the environmental factors related to water, temperature, and plants (Fig. 5, Supplementary Fig. 16, Supplementary Tables 2 and 3). Such linkages were greatly changed by warming (Fig. 5, Supplementary Fig. 16).

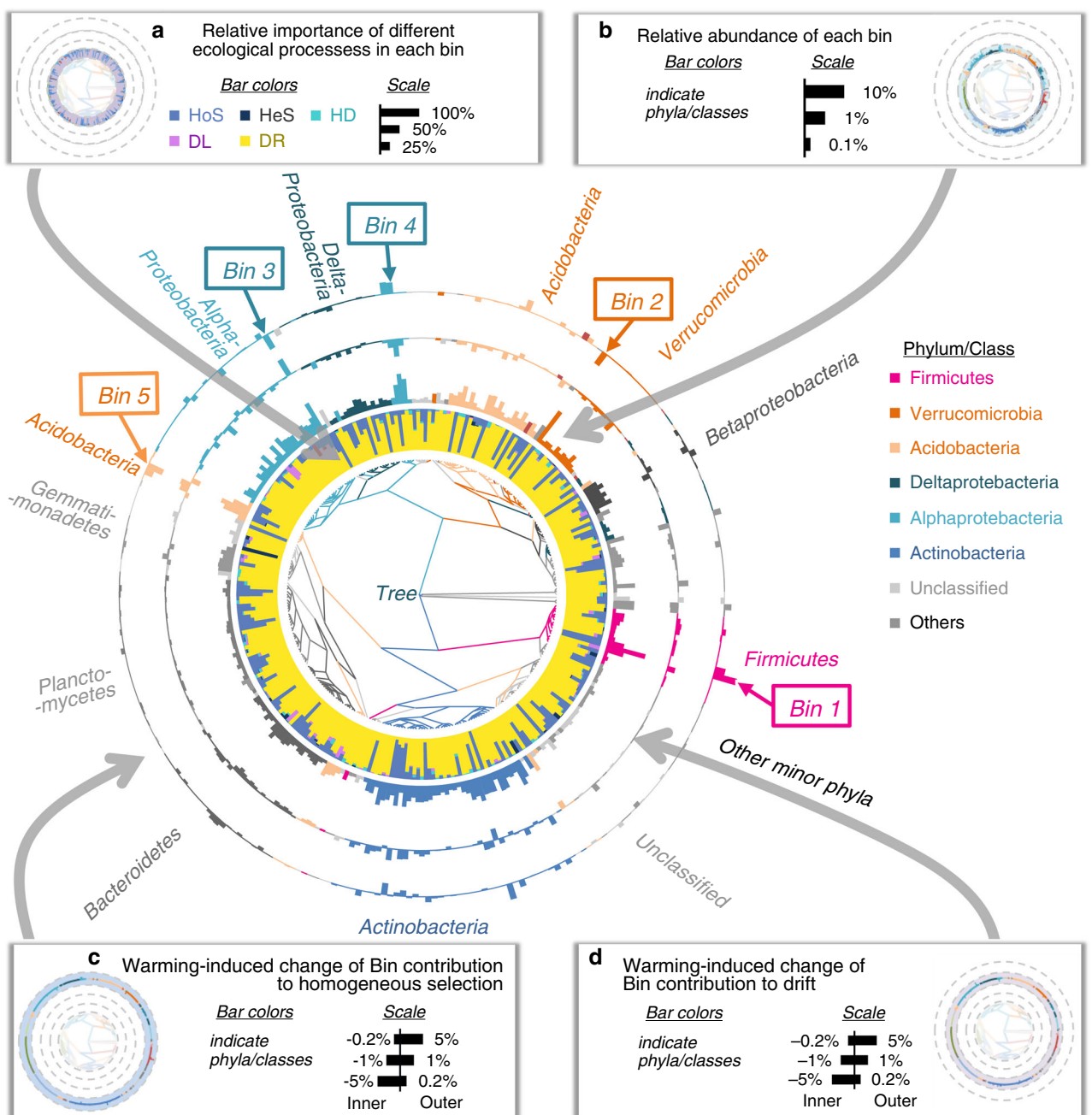

**Fig. 4 Variations of ecological processes across different phylogenetic groups.** Phylogenetic tree was displayed at the center. **a** Relative importance of different ecological processes in each bin (stacked bars in first annulus). Light livid bars, homogeneous selection (HoS); dark blue, heterogeneous selection (HeS); turquoise, homogenizing dispersal (HD); violet, dispersal limitation (DL); yellow, drift (DR). **b** Relative abundance of each bin (2rd annulus). **c** Warming-induced change in bin contribution to homogeneous selection (4th annulus) and **d** drift (third annulus) in later 3 years, where positive (outward bar) and negative (inward bar) represented increase and decrease by warming, respectively. Only the top 300 bins were shown in this figure, accounting for a total relative abundance of 95%. Bin 1 to Bin 5 were the five most abundant bins. Source data are provided as a Source Data file.

Homogeneous selection under control showed the strongest correlations with the C-4 biomass difference and plant richness (Mantel $R^2 > 0.33$, $p < 0.01$, Fig. 5a, Supplementary Table 2), and had slightly lower correlations with drought, precipitation, and moisture ($R^2 > 0.3$, $p < 0.01$). By contrast, homogeneous selection had the highest correlations with drought and precipitation ($R^2 = 0.52$–$0.57$, $p < 0.1$) under warming, followed by C-4, total plant biomass, and soil temperature ($R^2 > 0.32$, $p < 0.1$) (Fig. 5a). Interestingly, the total plant biomass under warming and the difference of C-4 biomass under control had strong correlations with homogeneous selection when the effects of drought or any

other factors were controlled (partial Mantel, Supplementary Table 3). Considering potential significant correlations among multiple factors, MRM was further used to determine the contributions of different environmental factors to homogeneous selection. Our result showed that the MRM models were able to explain a large portion of the plot-wise variations of homogeneous selection under warming ($R^2 = 0.94$, $p < 0.001$, Fig. 5b) and control ($R^2 = 0.86$, $p < 0.001$, Fig. 5c). The most important variables explaining homogeneous selection were soil temperature in sampling month under warming (partial regression coefficient $b = 0.92$, $p < 0.001$, Fig. 5b), and the between-plot difference of

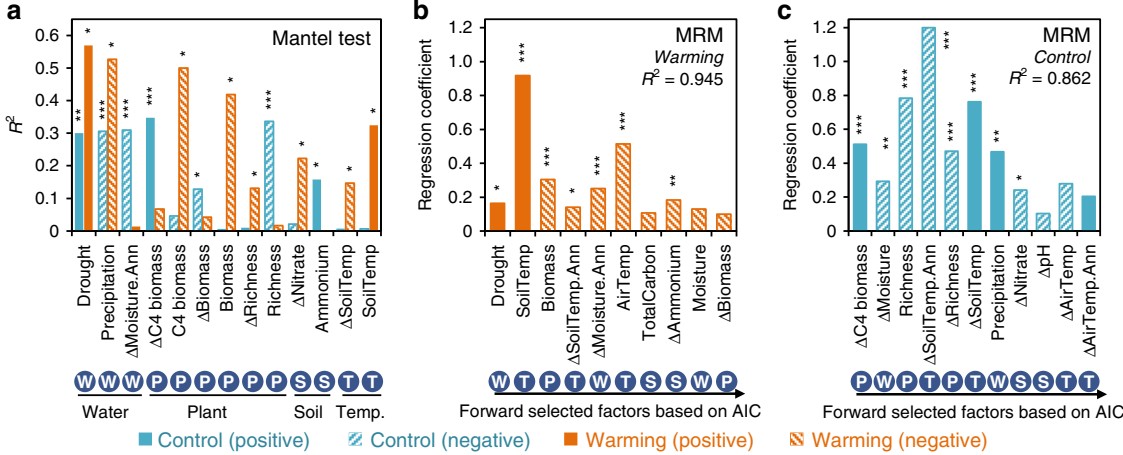

**Fig. 5 Effects of environmental factors on homogeneous selection. a** Correlations under warming (orange) and control (aqua) based on Mantel test. This figure showed only the factors with significant correlations. See Supplementary Table 2 for other factors. **b** Multiple Regression on distance Matrix (MRM) under warming. **c** MRM under control. $R^2$, coefficient of determination based on the best model from Mantel analysis (see Supplementary Table 2 for details). The correlation was determined based on the difference (with a triangle before the name) or the mean (without triangle) of a factor between each pair of samples. Factors marked with '.Ann' are annual means, while the remaining were measured in the sampling month. Livid circle under each factor name indicate the factor related to water (W), plant (P), soil properties (S), or temperature (T). Significance was expressed as ***$p < 0.01$; **$p < 0.05$; *$p < 0.1$ (see Source Data file for exact $p$ values). Temp, temperature. Source data are provided as a Source Data file.

annual soil temperature under controls ($b = -1.2$, $p < 0.001$, Fig. 5c). Meanwhile, some water-related and plant-related variables also showed significant, although smaller, effects ($|b| > 0.16$, $p < 0.1$) on homogeneous selection, including total plant biomass, drought, and moisture under warming (Fig. 5b), and plant richness, C-4 biomass, precipitation, and moisture in controls (Fig. 5c). These results indicated soil temperature, plant productivity, and drought were more effective in explaining the plot-wise variation of homogeneous selection under warming.

Drift had similar relationships to various environmental factors as homogeneous selection, but with opposite direction, based on Mantel (Supplementary Fig. 16a, Supplementary Table 2) and partial Mantel (Supplementary Table 3) tests. Under control condition, MRM models showed that soil total carbon and nitrogen, various plant biomass differences, and precipitation, had relatively strong effects on drift ($|b| > 0.36$, $p < 0.06$, Supplementary Fig. 16c). Under warming, the three most important variables shaping drift included total plant biomass, soil temperature, and drought ($|b| = 0.34–0.66$, $p < 0.005$, Supplementary Fig. 16b). Collectively, these results suggested that the environmental factors shaping homogeneous selection and drift were more similar under warming than control.

## Discussion

Disentangling ecological drivers controlling community assembly is crucial but difficult in ecology, especially in microbial ecology — mainly due to the huge diversity of microorganisms and the difficulties in their detection and quantification[1,3,17,18,36]. Although metagenomics and associated technologies have revolutionized microbial ecology research[1], a great challenge is how to use such massive data to address compelling ecological questions such as community assembly mechanisms. Thus, in this study, we developed a framework, iCAMP, to quantify the relative importance of different ecological processes underlying community diversity and dynamics based on individual phylogenetic groups (bins) rather than the entire community. Various analyses demonstrated that iCAMP improved performance substantially with higher precision, sensitivity, specificity, and accuracy compared with previous approaches. Also, the results from iCAMP

indicated important roles of stochastic processes in shaping the grassland soil microbial community with an average of ~60% stochasticity, which are consistent with those from various previous studies[3,8–14]. The developed framework would provide an effective and robust tool to quantify community assembly processes in microbial ecology towards mechanistic understanding of community diversity and succession. Given that our framework and simulation are general to high diversity communities rather than specific to microbes, it should also be applicable to plant and animal communities with high diversity.

To quantify selection, phylogeny-based approaches[20–22,24,25,37–39] require that the phylogenetic distances among taxa reflect their niche difference, i.e. there is phylogenetic signal or niche conservatism[3,21]. Although phylogenetic niche conservatism of microbial traits was reported[40,41], the signals were mostly at medium or low levels (i.e. close to or lower than Brownian Motion expectation)[42]. Fortunately, significant phylogenetic signals were frequently found within a short phylogenetic distance[25,38,40], which was employed by recent microbial studies[22,24,25,39], particularly in QPEN[19,20]. However, QPEN did not always perform well in inferring selection (e.g. in our simulated communities), possibly because it does not distinguish the differential influences of selection on distinct phylogenetic groups. By contrast, iCAMP partitions selection based on individual phylogenetic groups with essential phylogenetic signal embedded, and hence it can greatly improve quantitative performance (accuracy and precision > 0.7) with robustness to low phylogenetic signal across the tree and the complex assembly within individual bins. However, if competition predominates, closely related organisms may lead to strong competitive exclusion, which could disrupt phylogenetic conservatism, and thus significantly decrease the performance of iCAMP as showed in our simulation (Supplementary Fig. 11a–f). But given that the significant phylogenetic signal within short phylogenetic distances is widely observed[21,22,25,38], the disruption may not happen often under natural settings as theoretically predicted.

Unraveling the drivers controlling the responses of ecological communities to climate change is a critical topic in ecology and global change biology[34]. Several previous studies demonstrated that climate warming have significant impact on microbial diversity[43], structure[34], functional gene composition[44], and activities[44,45], but the underlying community assembly processes

were rarely examined. In our grassland site, experimental warming increased the soil temperature by ~3 °C (ref. [34]), thus it may gradually impose selective pressure as a deterministic force to decrease stochasticity as evident by our previous studies[34,43]. Here, iCAMP further revealed that warming gradually enhanced homogeneous selection and decreased drift in bacterial community assembly, and that the warming-induced selection was enhanced by drought and lower plant biomass, besides the soil temperature. These results are consistent with several previous studies showing that the effects of warming on microbial communities intertwines with precipitation/drought[46–49] and plant variables[50]. In addition, our results showed that the warming-enhanced homogeneous selection was mainly attributed to the positive responses of a group of Bacillales in Firmicutes, which are ubiquitous Gram-positive bacteria in nature. Bacillales are endospore-forming and can remain in the dormant state for years[51]. These traits could offer their competitive advantages under selective pressure from increased temperature and drying.

Although iCAMP has better performance over traditional approaches and provided valuable insights into the ecological processes governing the responses of grassland soil microbial communities to climate warming, there are still some limitations. For instance, one of the fundamental processes, diversification, is important to govern community assembly[15,17,18], but it is not explicitly accounted for in iCAMP. Diversification is still embedded with drift, weak selection, and/or weak dispersal in the 'drift' part of this framework. Given the lower performance of iCAMP when competition predominates selection, environmental filtering and biotic interactions need to be differentiated from each other in selection. In addition, although built on within-bin beta diversity, iCAMP should generally be able to capture important cross-bin selection (Supplementary Note 3). However, iCAMP might underestimate selection when cross-bin selection does not lead to detectable within-bin difference. Thus, further developments are needed by incorporating null model of evolution to infer the relative importance of diversification, and by integrating functional traits (genes) and network approaches with iCAMP to disentangle biotic interactions from abiotic filtering and capture special cross-bin selection.

## Methods

**Procedure of iCAMP**. Conceptually, selection under homogeneous abiotic and biotic conditions in space and time is referred to as constant selection[16] or homogeneous selection[20], by which low phylogenetic compositional variations or turnovers are expected. By contrast, selection under heterogeneous conditions leads to high phylogenetic compositional variations, which is referred to as variable selection[16,20] or heterogeneous selection[3]. Similarly, dispersal is also divided into two categories[19,20] — homogenizing dispersal and dispersal limitation. The former refers to the situation that high dispersal rate can homogenize communities and hence lead to little taxonomic compositional variations, whereas the later signifies the circumstance that low dispersal rates could increase community taxonomic variations. When neither selection nor dispersal is dominated, community assembly is governed by drift, diversification, weak selection and/or weak dispersal, which is referred to be 'undominated'[20] or simply designated as 'drift'[19].

To quantify these processes, iCAMP includes three major steps (Fig. 1). The first step is phylogenetic binning (Supplementary Figs. 1a and 3). Three binning algorithms were compared. One is based on the distance to abundant taxa (Supplementary Fig. 3a). The most abundant (i.e. the highest mean relative abundance in the regional pool) taxon is designated as the centroid taxon of the first bin. All taxa with distances to the centroid taxon less than the phylogenetic signal threshold, $d_s$, are assigned to this bin. The next bin is generated from the rest taxa in the same way. Consequently, a series of bins are generated with strict radiuses less than $d_s$, so-called strict bins. However, some strict bins may have too few taxa to provide enough statistical power for further analysis. Thus, each small bin is merged into its nearest-neighbor bin until all bins reach the minimal size requirement, $n_{min}$. The second algorithm is based on pairwise distances (Supplementary Fig. 3b). The first bin consists the most abundant taxon, and all other taxa among which all pairwise distances are lower than $d_s$. The second bin includes the next most abundant taxon among the remaining taxa. This procedure continues until all taxa are classified into different bins. To ensure each bin have enough size (≥$n_{min}$), a small bin less than $n_{min}$ is merged into the nearest neighbor

until all bins reach the minimal requirement $n_{min}$. The third algorithm is based on phylogenetic tree (Supplementary Fig. 3c). The phylogenetic tree is truncated at a certain phylogenetic distance (as short as necessary) to the root, by which all the rest connections between tips (taxa) are lower than the threshold $d_s$. The taxa derived from the same ancestor after the truncating point are grouped to the same strict bin. Then, each small bin is merged into the bin with the nearest relatives. This procedure is repeated until all merged bins have enough taxa (≥$n_{min}$). Although not used in this study, another option is also provided in our tool to omit small bins when they are negligible. However, all binning algorithms require a reliable phylogenetic tree, which might be difficult to construct for highly divergent marker genes such as ITS. In this case, certain special phylogenetic tree construction approaches (e.g. hybrid-gene[52] or constrained phylogenetic tree construction[53]) should be considered.

The objective of phylogenetic binning is to obtain adequate within-bin phylogenetic signal. To evaluate phylogenetic signal within each bin, the correlation between the pairwise phylogenetic distances and niche preference differences were analyzed by Mantel tests, where niche preference means the niche leading to optimum fitness (or relative fitness reflected by relative abundance) of a taxon. The bins with Pearson correlation coefficient $R > 0.1$ and $p < 0.05$ (one tail) are considered as bins with significant phylogenetic signal. In simulated communities, the niche preference difference between two taxa is treated as the key trait value difference. For empirical data, a practical index, i.e., niche value, is estimated as the relative-abundance-weighted mean of an environmental factor for each taxon[21]. For instance, if OTU1 has relative abundances of 10%, 20%, and 10% in three samples under the temperature of 10, 20, and 30 °C, respectively, the temperature niche value of OTU1 is $(10 \times 10\% + 20 \times 20\% + 3 \times 10\%)/(10\% + 20\% + 10\%) = 20$ °C. Then, the difference of niche values between taxa reflects niche difference, which are used for phylogenetic signal estimation. An optimized $n_{min}$ should give the highest number of bins with significant phylogenetic signal and relatively high average correlation coefficient (average $R$) within bins. Accordingly, the optimized $n_{min}$ is identified as 24 for simulated datasets and 12 for the empirical data. The index 'niche value' works under the assumption that relative abundances can represent the fitness of a taxon and the available environmental factors can measure the niche profile. Otherwise, an alternative way should be considered, i.e. choose the $n_{min}$ value that makes the estimated relative importance of stochastic processes close to the stochasticity assessed by referable approaches (e.g. pNST).

The second step is the null model analysis within each bin shown in Supplementary Fig. 1b. Accordingly, an operator is defined to count whether a bin is governed by a process. The operator can be calculated from βNRI and RC as Eqs. (1)–(10). Another significance testing index directly based on null model distribution is also provided in Supplementary Note 1.

$$W_{\text{HeS}uvk} = \begin{cases} 1 & \beta\text{NRI}_{uvk} > 1.96 \\ 0 & \text{else} \end{cases}, \tag{1}$$

$$W_{\text{HoS}uvk} = \begin{cases} 1 & \beta\text{NRI}_{uvk} < -1.96 \\ 0 & \text{else} \end{cases}, \tag{2}$$

$$W_{\text{DL}uvk} = \begin{cases} 1 & |\beta\text{NRI}_{uvk}| \leq 1.96 \text{ and } \text{RC}_{uvk} > 0.95 \\ 0 & \text{else} \end{cases}, \tag{3}$$

$$W_{\text{HD}uvk} = \begin{cases} 1 & |\beta\text{NRI}_{uvk}| \leq 1.96 \text{ and } \text{RC}_{uvk} < -0.95 \\ 0 & \text{else} \end{cases}, \tag{4}$$

$$W_{\text{DR}uvk} = \begin{cases} 1 & |\beta\text{NRI}_{uvk}| \leq 1.96 \text{ and } |\text{RC}_{uvk}| \leq 0.95 \\ 0 & \text{else} \end{cases}, \tag{5}$$

$$\beta\text{NRI}_{uvk} = \frac{\beta\text{MPD}_{uvk} - \overline{\beta\text{MPD}_{\text{null}uvk}}}{\text{Sd}(\beta\text{MPD}_{\text{null}uvk})}, \tag{6}$$

$$\text{RC}_{uvk} = 2 \frac{\sum_{n_r=1}^{N_r} \delta_{uvk}^{(n_r)}}{N_r} - 1, \tag{7}$$

$$\delta_{uvk}^{(n_r)} = \begin{cases} 1 & \text{BC}_{\text{null}uvk}^{(n_r)} < \text{BC}_{uvk} \\ 0.5 & \text{BC}_{\text{null}uvk}^{(n_r)} = \text{BC}_{uvk} \\ 0 & \text{BC}_{\text{null}uvk}^{(n_r)} > \text{BC}_{uvk} \end{cases}, \tag{8}$$

$$\beta\text{MPD}_{uvk} = \frac{\sum_i^{S_k} \sum_j^{S_k} f_{iu} f_{jv} d_{ij}}{\sum_i^{S_k} \sum_j^{S_k} f_{iu} f_{jv}}, \tag{9}$$

$$\text{BC}_{uvk} = \frac{\sum_i^{S_k} |x_{iu} - x_{iv}|}{\sum_i^{S_k} (x_{iu} + x_{iv})}, \tag{10}$$

where '$W_{\text{HeS}uvk}$' is operator for heterogeneous selection, to count whether the turnover of the $k$th phylogenetic bin (Bin $k$) between community $u$ and $v$ governed by heterogeneous selection. $W_{\text{HoS}uvk}$, $W_{\text{DL}uvk}$, $W_{\text{HD}uvk}$, and $W_{\text{DR}uvk}$ are analogous

operators for homogeneous selection, dispersal limitation, homogenizing dispersal, or 'drift', respectively. '$\beta NRI_{uvk}$' is bNRI of Bin $k$ between community $u$ and $v$. '$\beta MPD_{uvk}$' is beta mean pairwise distance of Bin $k$ between communities $u$ and $v$, and '$\beta MPD_{null_{uvk}}$' is the $\beta MPD$ of the null communities randomized according to a null model. 'Sd' is standard deviation. '$RC_{uvk}$' is modified RC. '$N_r$' is total randomization times, usually 1000 times. '$\delta_{uvk}^{(n_r)}$' is an operator to calculate RC value. '$BC_{uvk}$' is Bray–Curtis dissimilarity index. '$BC_{null_{uvk}}$' is Bray–Curtis dissimilarity of Bin $k$ between null communities $u$ and $v$ of the $n_r$th time randomization according to a null model. '$f_{iu}$' and '$f_{jv}$' represent relative abundance of taxon $i$ in community $u$ or taxon $j$ in community $v$, respectively. '$S_k$' represents taxa number in Bin $k$. '$x_{iu}$' and '$x_{iv}$' are abundance of taxon $i$ in communities $u$ and $v$, respectively. For microbial data from sequencing, it is usually difficult to get accurate estimation of absolute abundances of taxa in a community, thus relative abundances can be used to calculate Bray–Curtis index as a common practice.

The null model algorithm for phylogenetic metrics is 'taxa shuffle'[21,37], which randomizes the taxa across the tips of the phylogenetic tree, and thus it randomizes the phylogenetic relationship among taxa. The null model algorithm for taxonomic metric is the one constraining occurrence frequency of each taxon proportional to observed and richness in each sample fixed to observed[19,54]. The null model algorithm results heavily depend on the selection of the regional pool, within which randomization is implemented[54]. Thus, the algorithms randomizing taxa within each bin and across all bins were compared in iCAMP analysis for the simulated communities. No matter whether the randomization is within or across bins, the beta diversity metrics are calculated in each bin as defined in Eqs. (6)–(10).

Null model analysis is most computational resource — and time-consuming, which largely depends on the times of randomization and taxa number. But decreasing randomization times or taxa number can reduce reproducibility of the null model analysis. Considering that most reported null model analyses used 1000-time randomization, iCAMP were performed for simulated data with randomization times ranging from 25 to 5000 and repeated 12 times with each number of randomization times. The results from 60,000-time randomization served as a standard for evaluation. In addition, three methods for reducing taxa number were tested. The method 'rarefaction' means to randomly draw the same number of individuals (sequences) from each sample and reduce the taxa number. The method 'average abundance trimming' ranks all taxa from abundant to rare according to their average relative abundances across all samples and only keeps the taxa before a certain rank. The method 'cumulative abundance trimming' ranks taxa in each sample from abundant to rare, then only keeps the abundant taxa in each sample so that every sample has the same cumulative abundance. The iCAMP results from the three methods were compared to that from the original simulated communities.

The third step of iCAMP is to integrate the results of different bins to assess the relative importance of each process (Supplementary Fig. 1c–f). Defining neutrality at individual level has been proved a key to successfully develop the unified neutral theory[6]. Therefore, the relative importance of a process can be quantitatively measured as abundance-weighted percentage for each bin (Eq. (11)) or the entire communities (Eqs. (12) and (13)). Qualitatively, for each pairwise comparison between communities (samples), the process with higher relative importance than other processes is regarded as the dominant process.

$$P_{\tau k} = \frac{\sum_{uv}^{m} \frac{f_{uk}+f_{vk}}{2} W_{\tau uvk}}{\sum_{uv}^{m} \frac{f_{uk}+f_{vk}}{2}}, \quad (11)$$

$$P_{\tau uv} = \sum_{k=1}^{K} \frac{f_{uk}+f_{vk}}{2} W_{\tau uvk}, \quad (12)$$

$$P_{\tau} = \frac{\sum_{uv}^{m} P_{\tau uv}}{m} = \sum_{k=1}^{K} f_k P_{\tau k}, \quad (13)$$

where '$P_{\tau k}$' is relative importance of the $\tau$th ecological process in governing the turnovers of Bin $k$ among a group of communities (e.g. samples within a treatment, a region, etc.; Supplementary Fig. 1d) or between a pair of groups (e.g. between treatment and control, which can be enabled by set 'between.group' as TRUE for functions 'icamp.bins' and 'icamp.boot' in iCAMP package). '$P_{\tau uv}$' is relative importance of the $\tau$th ecological process in governing the turnover between communities $u$ and $v$ (Supplementary Fig. 1c). '$P_{\tau}$' is relative importance of the $\tau$th ecological process in governing the turnovers among a group of communities (Supplementary Fig. 1c) or between a pair of groups. Thus, $P_{\tau}$ can be $P_{HeS}$, $P_{HoS}$, $P_{DL}$, $P_{HD}$, or $P_{DR}$ for heterogeneous selection, homogeneous selection, dispersal limitation, homogenizing dispersal, or 'drift', respectively. '$f_{uk}$' and '$f_{vk}$' are total relative abundance of Bin $k$ in community $u$ and community $v$, respectively. '$W_{\tau uvk}$' is operator counting whether the $k$th bin is governed by the $\tau$th ecological process, including $W_{HeSuvk}$, $W_{HoSuvk}$, $W_{DLuvk}$, $W_{HDuvk}$, and $W_{DRuvk}$ (Eqs. (1)–(5)). '$K$' is total number of bins. '$m$' is number of pairwise comparisons in a group of communities (e.g. within a treatment) or between a pair of groups (e.g. between treatments). '$f_k$' is average relative abundance of Bin $k$ in the group of communities.

As shown in Eq. (13), the relative importance of each process $P_{\tau}$ is the sum of the terms $f_k P_{\tau k}$, by which we can define the contribution of different bins to $P_{\tau}$

(Eqs. (14) and (15)).

$$BP_{\tau k} = f_k P_{\tau k} = \frac{\sum_{uv}^{m} \frac{f_{uk}+f_{vk}}{2} W_{\tau uvk}}{m}, \quad (14)$$

$$BRP_{\tau k} = \frac{BP_{\tau k}}{P_{\tau}} = \frac{\sum_{uv}^{m} \frac{f_{uk}+f_{vk}}{2} W_{\tau uvk}}{\sum_{k=1}^{K} \sum_{uv}^{m} \frac{f_{uk}+f_{vk}}{2} W_{\tau uvk}}, \quad (15)$$

where '$BP_{\tau k}$' is Bin contribution to Process, measuring the contribution of Bin $k$ to the relative importance of $\tau$th ecological process in the assembly of a group of communities (Supplementary Fig. 1e). '$BRP_{\tau k}$' is Bin Relative contribution to Process, measuring the relative contribution of Bin $k$ to the $\tau$th ecological process (Supplementary Fig. 1f).

**Simulation model.** In the simulation model (Supplementary Fig. 2), all samples are from the same region sharing the same metacommunity (the regional species pool) with 20 million individuals. The relative abundances of species in metacommunity are simulated using metacommunity zero-sum multinomial distribution model (mZSM) derived from Hubbell's Unified Neutral Theory Model[55], using R package 'sads' (version 0.4.2)[56] with $J = 2 \times 10^7$ and $\theta = 5000$. The whole region has two separated islands of A and B (Supplementary Fig. 2a). For species controlled by dispersal, migration is unlimited within each island but nearly impossible between islands. Each island has two plots: plot LA and HA at island A, and plot LB and HB at island B. The two plots at the same island are under distinct environments. The environment variable is as low as 0.05 in the north plots at each island (LA and LB), but as high as 0.95 in the south plots (HA and HB), which is a critical setting for species under niche selection. At each plot, six local communities are simulated and sampled as biological replicates. Each local community contains 20,000 individuals of 100 species.

A phylogenetic tree was retrieved from a previous publication[20], which simulated evolution from a single ancestor to the equilibrium between speciation and extinction and generated a tree with 1140 species. A trait defining the optimal environment of each species ($E_i$) evolves along the phylogenetic tree with a certain phylogenetic signal. We simulated three pools of species as three scenarios to explore the performance of iCAMP under distinct levels of phylogenetic signals. (i) The low-phylogenetic-signal pool was generated using Stegen's evolution model[20]. The Blomberg's $K$ value is as low as 0.15, close to the mean $K$ value of 91 continuous prokaryotic traits[42]. The phylogenetic signal is low if counting the phylogenetic distance across the whole tree. However, the trait still shows significant phylogenetic signal within a short phylogenetic distance[20], in accordance with general observations in microbial communities in various environments[19,38]. (ii) The medium-phylogenetic-signal pool was generated by simulating the trait according to Brownian motion model, using the function 'fastBM' in R package 'phytools' (version 0.6–99)[57] with an ancestral state of 0.5, an instantaneous variance of Brownian process of 0.25, and the boundary from 0 to 1. The final $K$ value is 0.9, close to the mean phylogenetic signal level of 899 prokaryotic binary traits[42]. (iii) The high-phylogenetic-signal pool was simulated according to Blomberg's ACDC model[58] with a $g$ value of 2000. The final $K$ value is as high as 5.5, close to the highest phylogenetic signal of prokaryotic traits to date[42].

For each scenario, we simulated 15 situations with different levels of expected relative importance of various processes (Supplementary Fig. 2b). The situations can be classified into two types. In the first type, all species under each situation are governed by the same kind of processes, i.e. pure selection, or dispersal, or drift. In each of the other situations, species in the regional pool are assigned to different types controlled by various processes. Once a species is assigned to be controlled by selection or dispersal rather than drift, its nearest relatives within $d_s$ will also be assigned to the same type of processes considering the phylogenetic signal of traits. Species controlled by each type of processes are simulated as below. (i) To simulate strong selection due to abiotic filtering without stochasticity, the relative abundance of each species is determined by the difference between the environment variable and their trait values (optimal environment), following a Gaussian function (Eq. (16), Supplementary Fig. 2d).

$$A_{ij} \propto \exp\left[-\frac{(EV_j - E_i)^2}{2\sigma_E^2}\right], \quad (16)$$

where '$A_{ij}$' is abundance of species $i$ in local community $j$. '$EV_j$' is the value of the key environmental variable in local community $j$, which is 0.05 in the north plots (LA and LB) and 0.95 in the south plots (HA and HB). '$E_i$' is the optimum environment of species $i$. '$\sigma_E$' is the standard deviation, which is 0.015. Consequently, the turnovers of these species under the same environment (i.e. within north plots, or within south plots) are solely governed by homogeneous selection, and those between distinct environments (i.e. between north and south plots) are governed by heterogeneous selection.

(i) To simulate competition without stochasticity, a geometric series model[59] was modified to consider stronger competition between species with similar niche preference[37]. Competitive species in a local community are ranked from the strongest competitor to the weakest with their relative abundances proportional to 0.5, $0.5^2$, $0.5^3$, …, $0.5^h$, …. The strongest competitor is randomly selected from species with the best fitness, i.e. from the top 10 species with the lowest $|EV_u - E_i|$.

Then, the next competitor is the one with the largest niche difference with prior competitor(s) in the rank, based on abundance-weighted Euclidean trait distance[37] to previous competitor(s) (Eq. (17)). The total relative abundance of species controlled by competition is determined as the designated ratio of competition in selection multiplied by the designated ratio of selection in a simulated situation. The turnovers of these species are defined as governed by selection, without distinguishing between homogeneous and heterogenous selections.

$$nd_{hi} = \sqrt{\sum_{j=1, i>j}^{h-1} 0.5^j (E_i - E_j)^2}, \qquad (17)$$

where '$nd_{hi}$' is the index to assess niche difference between species $i$ and $(h-1)$ prior competitors in the rank. The species with the highest $nd_{hi}$ will be the $h$th competitor in the rank, and assigned relative abundance proportional to $0.5^h$. '$E_i$' is the optimum environment of species $i$ which is not included in the $(h-1)$ prior competitors. '$E_j$' is the optimum environment of species $j$ which is the $j$th prior competitor with relative abundance proportional to $0.5^j$.

(ii) To simulate extreme dispersal without selection, we modified Sloan's simulation model[60] which was derived from Hubbell's neutral theory model (Supplementary Fig. 2e). Each island has a unique species pool, simulated as a local community under the regional metacommunity following neutral theory model but with a relatively low dispersal rate ($m_1 = 0.01$). However, the unique species pools of the two islands are constrained to have no overlapped species, regarding extreme dispersal limitation between the two islands. Then, the local communities in each island are simulated as governed by neutral dispersal from both the regional metacommunity with a low rate ($m_1 = 0.01$) and the unique species pool of the island with a high rate ($m_2 = 0.99$). It means 99% of dead individuals in a local community are replaced by species from the small island–unique species pool at each time step. Therefore, all the turnovers within an island are governed by homogenizing dispersal, and those between islands are controlled by dispersal limitation.

(iii) Drift is simulated as neutral stochastic processes at a moderate dispersal rate rather than limited or strong dispersal. To simulate drift, all local communities are generated under neutral dispersal from the regional metacommunity with a medium rate ($m_1 = 0.5$, Supplementary Fig. 2c). Since 50% of dead individuals are replaced by species randomly drawing from a relatively large regional pool, all the turnovers among local communities are neither affected by homogenizing dispersal nor under dispersal limitation.

Under each situation, the dataset of the 24 local communities is simulated as a combination of species governed by different ecological processes, with ratios defined by the situation setting (Supplementary Table 1, Supplementary Fig. 2b). To simulate complex assembly of bins, the species pool is divided into bins with different bin size limitation ($n_{min} = 3, 6, 12, 24, 48$) and phylogenetic distance cutoff ($d_s = 0.1, 0.2, 0.4$), and each bin is simulated as controlled by a certain process. Then, as iCAMP analysis still uses $n_{min} = 24$ and $d_s = 0.2$, some estimated bins can have members governed by different processes in the same bin. For each turnover between a pair of local communities, the mean relative abundance of species governed by a process defines the expected relative importance of the process (Eq. (18)). The process with the highest relative importance is the expected dominant process of the turnover. Since dispersal and drift are simulated as pure stochastic processes, the expected stochasticity is defined as the sum of expected relative importance of homogenizing dispersal, dispersal limitation, and drift (Supplementary Table 1).

$$EP_{\tau uv} = \sum_{i=1}^{K} \frac{f_{uk} + f_{vk}}{2} \omega_{\tau uvk}, \qquad (18)$$

where '$EP_{\tau uv}$' is the expected relative importance of the $\tau$th ecological process in community turnover between samples $u$ and $v$. '$f_{uk}$' is total relative abundance of Bin $k$ in community $u$. '$f_{vk}$' is total relative abundance of Bin $k$ in community $v$. '$\omega_{\tau uvk}$' is operator, equal to 1 if the turnover of the $k$th bin between communities $u$ and $v$ is governed by the $\tau$th ecological process, and equal to 0 if not.

We simulated three scenarios with different levels of phylogenetic signal, 15 situations per scenario with 1 dataset per situation, thus a total of 45 datasets. In each dataset, we applied both QPEN and iCAMP to estimate the relative importance of different processes (quantitative estimation) and the dominant process (qualitative estimation). QPEN cannot assess relative importance of processes for each turnover, but can estimate their relative importance as the percentage of turnovers governed by the process in all turnovers within a plot (e.g. plot HA) or between a pair of plots (e.g. plot HA vs. HB). Then, the ecological stochasticity of community assembly can be quantified as the relative importance of stochastic processes (i.e. homogenizing dispersal, dispersal limitation, and drift) based on QPEN and iCAMP, respectively. For comparison, the ecological stochasticity in each dataset is also estimated with NP[61], tNST[33], and pNST[33,34].

The performance of quantitative estimation is evaluated by accuracy (Eq. (19)) and precision coefficients (Eq. (20)) derived from concordance correlation coefficient (CCC)[62]. The performance of qualitative estimation is assessed with respect to accuracy, precision, sensitivity, and specificity by counting the true and

false positive/negative results (Eqs. (21)–(24)).

$$qACC = \frac{2\sigma_x \sigma_y}{\sigma_x^2 + \sigma_y^2 + (\mu_x - \mu_y)^2}, \qquad (19)$$

$$qPRC = \frac{\sigma_{yx}}{\sigma_x \sigma_y}, \qquad (20)$$

where 'qACC' and 'qPRC' are quantitative accuracy and precision, respectively. '$\sigma_{yx}$' is covariance of $x$ and $y$. In our study, $x$ and $y$ are the expected and estimated stochasticity or relative importance of a process, respectively. '$\sigma_x^2$' and '$\sigma_y^2$' are variance of $x$ and $y$, respectively. '$\mu_x$' and '$\mu_y$' are mean of $x$ and $y$, respectively.

$$ACC = \frac{TP + TN}{TP + TN + FP + FN}, \qquad (21)$$

$$PRC = \frac{TP}{TP + FP}, \qquad (22)$$

$$SST = \frac{TP}{TP + FN}, \qquad (23)$$

$$SPC = \frac{TN}{TN + FP}. \qquad (24)$$

In the qualitative performance indexes, 'ACC' is accuracy; 'PRC' is precision measured as positive predictive value; 'SST' is sensitivity measured as true positive rate; 'SPC' is specificity measured as true negative rate. 'TP' is true positive number. A true positive for a process means a turnover is correctly identified as dominated by this process. Overall true positive of a method is calculated as the sum of true positive numbers of all processes. 'TN' is true negative number. A true negative for a process means a turnover is correctly identified as not dominated by this process. Overall true negative is calculated as the sum of true negative numbers of all processes. 'FP' is false positive number. A false positive for a process means a turnover is incorrectly identified as dominated by this process. Overall false positive is calculated as the sum of false positive numbers of all processes. 'FN' is false negative number. A false negative for a process means a turnover is incorrectly identified as not dominated by this process. Overall false negative is calculated as the sum of false negative numbers of all processes.

For example, a turnover is in fact dominated by drift. If the estimated dominating process is drift, this is a true positive for drift, and a true negative for other processes. If the estimated dominating process is dispersal limitation, this is a false positive for dispersal limitation and a false negative for drift, but a true negative for other processes.

**Experimental data and analyses.** We applied iCAMP to an empirical dataset from our previous study[34], with sequencing data available in the NCBI Sequence Read Archive under project no. PRJNA331185. Briefly, the grassland site is located at the Kessler Atmospheric and Ecological Field Station (KAEFS) in the US Great Plains in McClain County, Oklahoma (34°59′N, 97°31′W)[34]. The field site experiment was established in July of 2009. Surface soil temperature in warming plots (2.5 m × 1.75 m each) is increased to 2–3 °C higher than the controls by utilizing infrared radiator (Kalglo Electronics, Bath, PA, USA). Surface (0–15 cm) soil samples were taken annually from four warming and four control plots. A total of 40 samples over 5 years after warming (2010–2014) were analyzed in this study. Soil DNA was extracted by from 1.5 g of soil by freeze-grinding and SDS-based lysis[63] and purified with a MoBio PowerSoil DNA isolation kit (MoBio Laboratories). Then the V4 region of 16S rRNA gene was analyzed by amplicon sequencing on Illumina MiSeq[34], using the primers 515F (5′-GTGCCAGCMGCCG CGGTAA-3′) and 806R (5′-GGACTACHVGGGTWTCTAAT-3′). Sequencing results were analyzed with our pipeline (http://zhoulab5.rccc.ou.edu:8080)[34] built on the Galaxy platform (version 17.01)[64] and OTUs were generated by UPARSE[65] at 97% identity. Soil properties were analyzed using a dry combustion C and N analyzer (LECO), a Lachat 8000 flow-injection analyzer (Lachat), pH meter, a portable time domain reflectometer (Soil Moisture Equipment Corp.), and constantan–copper thermocouples with CR10x data logger (Campbell Scientific)[34]. Plant biomass was measured with a modified pin-touch method and the plant richness was based on identification of all species in each plot[34]. The drought index is calculated as additive inverse of standardized precipitation–evapotranspiration index (SPEI) retrieved from SPEIbase[66].

**Statistical analyses.** The significance of difference for each evaluation index (e.g. qualitative accuracy, precision, sensitivity, etc.) between different methods was calculated by bootstrapping for 1000 times (one-side test). To assess the effects of warming on ecological processes, the standardized effect size (Cohen's $d$) was calculated as the difference of means between warming and controls divided by the combined standard deviation, and the magnitude of effect is defined as large ($|d| > 0.8$), medium ($0.5 < |d| \leq 0.8$), small ($0.2 < |d| \leq 0.5$), and negligible ($|d| \leq 0.2$) according to Cohen's $d$[67]. NST[33], NP[61] and QPEN[19,20] were applied to the dataset. The significance of difference in stochasticity or relative importance of ecological processes between warming and control was calculated by permutational $t$ test

(1000 times). The empirical study only investigated within-treatment spatial turnovers at each time point.

For correlation test between each process and various measured factors, we applied Mantel test, partial Mantel[68], and MRM[69] with constrained permutation considering repeated measures design of the experiment. For Mantel test, both linear model and general linear model with a logit link function and a 'quasibinomial' distribution were tested, and the relative importance of each process and each factor were either log-transformed or not, to explore the best model. To log-transform a factor with zero or negative values, all its values were subtracted by the lowest value and the resulted zero values were replaced by 0.05 (i.e. $-3.00$ in natural log) of the minimum positive value before natural $-$log transformation. Partial Mantel was performed on factors with significant correlation with homogeneous selection or drift. For MRM, the factors were log-transformed and standardized to zero-mean and unit-variance, then the best model was forward selected based on Akaike information criterion (AIC). For each measurement (e.g. soil nitrate), both the variation (e.g. $|\text{Nitrate}_u - \text{Nitrate}_v|$, where $u$ and $v$ represent samples) and the mean (e.g. $[\text{Nitrate}_u + \text{Nitrate}_v]/2$) in each pair of samples were investigated for correlation with the relative importance of each process (e.g. $P_{\text{HoS}uv}$). All statistical analyses were implemented by R (version 3.5.3)[70]. All significance tests are two-side unless specified.

**Reporting summary**. Further information on research design is available in the Nature Research Reporting Summary linked to this article.

## Data availability

The sequencing data are available in the NCBI Sequence Read Archive under project no. PRJNA331185. The source data underlying Figs. 2–5 and Supplementary Figs. 3–17 are provided in the Source Data file. Other source data are all available from GitHub (https://github.com/DaliangNing/iCAMP1), such as OTU tables, phylogenetic trees, treatment information, etc. All other data are available from the authors upon reasonable request. Source data are provided with this paper.

## Code availability

Code is available as an open-source R package 'iCAMP', which can be downloaded from the Comprehensive R Archive Network (CRAN, https://cran.r-project.org/)[70]. iCAMP can also be implemented on a web-based pipeline (http://ieg3.rccc.ou.edu:8080) built on Galaxy platform (version 18.09)[64]. The R package and an example with detailed notes are also available in the Supplementary Code file. All custom scripts are available from GitHub (https://github.com/DaliangNing/iCAMP1). Source data are provided with this paper.

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

## Acknowledgements

The development of the theoretical framework was supported by the U.S. Department of Energy, Office of Science, Office of Biological and Environmental Research (DOE-BER) (DE-SC0014079, DE-SC0016247, and DE-SC0020163); also part of ENIGMA—Ecosystems and Networks Integrated with Genes and Molecular Assemblies (http://enigma.lbl.gov), a Scientific Focus Area Program at Lawrence Berkeley National Laboratory, supported by DOE-BER under contract number DE-AC02-05CH11231; and by the U.S. National Science Foundation (NSF) (EF-1065844). The experimental data was generated with the support from DOE-BER under award number DE-SC0010715.

## Author contributions

All authors contributed the intellectual input and assistance to this study and manuscript preparation. J.Z. conceived the research questions. D.N. and J.Z. developed the mathematical framework. M.Y., X.G., and X.Z. collected the empirical data. D.N. developed simulated communities and performed statistical analysis. J.Z. and D.N. wrote the paper with inputs from M.Y., L.W., Y.Z., X.G., Y.Y., A.P.A., and M.K.F.

## Competing interests

The authors declare no competing interests.
