## [Peer Review File · Nature Communications]

Reviewers' Comments:

Reviewer #1:

Remarks to the Author:

This article presents an interesting approach to study phylogenetic diversity pattern of microbial communities. While I am not overly familiar with diversity pattern analysis in microbial studies, I think the approach is new and looks well rounded. The inclusion of a simulation study is remarkable as most newly devised diversity pattern methodologies dispense themselves from doing such a validation work. Furthermore while the analysis is undeniably complex, a functioning R-package was provided which suggests that the method will be easily used by other microbial ecologists

This being said, I have two major concerns, one on the methodology and one on the substance of the analysis. At the moment, I feel that the empirical case study do not convince me that iCAMP is a superior analysis to QPEN.

Methodological concerns:

(1) While I may have missed it, I failed to find the test of the different null models combination (Figure S3). Phylogenetic beta diversity can be tested by both a within-bin randomization or a across-bin randomization, same for the taxonomic beta diversity. So four null models/indices combinations should be tested.

(2) When the main ecological process is assessed within bins, is it using the across-bin null model for the taxonomic beta-diversity (which I would deem inadequate for accessing the ecological process occurring within a given bin), or is it with the within-bin null model?

(3) In the empirical experiment, it seems that warming mostly induces a change in bin abundances rather than changing the ecological process that occur within bins. As stated by the authors, the increasing of homogenizing selection seem to be mainly driven by an increasing relative abundance of Firmicutes that are under homogenizing selection regardless of the temperature treatment. So, if the analysis suggests that the effect of warming plays a selective role between phylogenetic bins (i.e. by favouring Firmicutes over other clades), is it really relevant to analyse the effect of the warming treatment within bins?

Similarly, I am wondering what is the logic of not analysing the beta-diversities between the control and the treated communities, it seems that it would be an interesting way to access whether warming generates heterogeneous selection on microbial communities.

Form concern: I feel that the analysis is too layered to be easily grasped by the reader. I understand that the authors want to present a thorough analysis but even an attentive reader like me could feel lost. It is aggravated by the introduction-results-discussion-methods layout of Nature Communications that is not ideal to handle such a methodological paper.

I would recommend the author to trim the article and put (even) more analytical elements in the supplementary materials, especially all the ones are not essential to understand the methodology and grasp its value (good candidates would be the comparison of the algorithms hacking the trees to produce phylogenetic bins and the comparison of various diversity indices).

I recommend limiting the use of acronyms. While it is perfectly ok for algorithms and diversity index names, when it is about ecological processes (HoS etc), I find them unpleasant to read.

Various remarks:

l. 38: the most diverse group**s** - 'Microorganisms' are paraphyletic

l.40: 'are critical to our society' - they are critical to ecosystems in general really.

l.109 - 110: how were determined those thresholds?

L. 210: can you suggest why the

l. 254: Do you have an interpretation regarding the increase of Firmicutes abundances in the control plot for the first three years?

Figure 3 : pretty but incomprehensible. Barplots are misleading, rather see it on a 0-1 scale. I am not convinced that overlaying effect size and barplot helps the interpretation.

Figure 4 : Too many redundant colours, maybe a way to simplify the figure would be to class more bins into the 'other' category.

Fig S14: I would suggest to test the model $HoS \sim Drought + Plant\ cover$ to distinguish the two effects. I am suspecting that drought and plant cover are related. Partial mantel test could be used for instance.

Figure 5 : Similarly a multi factor model should maybe be preferred. All these factors are likely to be somewhat correlated. Mantel test extensions like multiple regressions on distance matrices could be used <https://link.springer.com/article/10.1007/s11258-006-9126-3> .

Reviewer #2:

Remarks to the Author:

This MS is on the development of a new way to examine community assembly processes for microbial community data. It is certainly a worthwhile field to study as the authors correctly point out that these processes which underlay almost all community dynamics are, so far, not accounted for in any meaningful way. I applaud the amount of work and detail that went into this MS. Further, besides a few places (see below), it is a very well-written MS. Despite the potential great advance this work may yield, there are a few major concerns I had. Primarily, these should be able to be addressed with better justification and some simulation work. Line by line notes are presented below. The major concerns I have are more that I take some issues with the assumptions made about how these taxa would behave that underlay how these assembly processes are derived. Namely, it is fair to assume that all members within a bin have the same niche-space and functional roles, thus is it fair to assume that all members of the bin would have the same partition of the assembly processes. As written, I would say no, and given that this is a MAJOR baseline assumption, this casts suspicion on how the rest of the data that results can be trusted. I recommend conducting a simulation exercise that will demonstrate that this underlying assumption (even if false) will not impact obtained results much. Also, is it fair to combine bins to meet the minimum size, even if these were originally binned separately (I am not convinced)? Further, I feel the authors use very strong language, almost aggressive language, when they compare their results against QPED, but provide no justification for such seemingly denegation. The authors may have there reasons but the reader is not privy to this. Finally, given line 88, can iCAMP be modified to run with gene data (each gene is an OTU)? It is implied that this is needed but it does not appear that iCAMP can.

Abstract:

Line 31: Change "on Bacillales" to 'on the Bacillales'

Line 31 -32: This sentence is awkward. What is strengthened exactly. The 'selection' or the impact of selection on community assembly. This just needs to be a bit more precise.

Line 33: Strengthen this closing sentence.

Introduction:

Line 42: What mechanisms are referred to here? This is a very open statement and the challenge of understanding various mechanisms is not constant.

Lines 42-43: Is NGS development still 'recent'? Lines exactly like this one have been used for the past decade.

Lines 44-45: Why?

Lines 50-51: I take issue with the presumption that these processes are stochastic. Stochasticity plays a large role of course, but there are many other factors that are not random and it is the combined action that drives the rates.

Line 51: Awkward phrase

Line 56: "Umbrella" seems an odd choice of word here.

Lines 62-64: Why?

Lines 64-80: I feel like there is too much detail here, do we need a full breakdown of operations definitions here? Particularly given that you later say that framework is limited, thus justifying your new framework.

Line 82-84: While there is no doubt that this is an advance, I do not find it fair to claim that this allowed quantification of assembly processes in microbes for the first time. Many examples exist where careful experimentation has allowed such quantification (even as early as the 80s). However, not with this statistical power, but please be careful to not overpromise novelty.

Lines 85-88: Is there not an argument to be made that if communities change, then lower level of organizations will change too, such as populations and gene complements? Of course I am not arguing that only communities matter, but to illustrate the importance for careful selection of words.

Line 91-92: Awkward phrasing.

One thing that is missing is the direct statement of what iCAMP compares. There is some confusing language that makes it uncertain if this compares all data from multiple samples or on a sample by sample basis. Give beta-NRI and RC, if must be across samples, but this should be explicitly stated.

Results:

Lines 106-107: Is it fair to assume that each bin (members of closely related organisms) should all behave similarly with respect to ecological processes. It is quite easy to contrive an example where even strain level variation (single terminal node depending on binning) would differ in their relative impacts to drift, selection etc. I would require some deep discussion and simulations to believe that this could not have an effect of process outcomes. This ignores competitive exclusion. Is niche conservatism by closely related organisms the only option? I would say no.

Lines 111-114: Traditionally, Raup-Crick measures are binary in nature and are the probabilistic

inferring if one selection of fossils is a random subset of another selection. 1) Is a binary matrix approach powerful enough to be used in fine scale parsing of processes, especially if this is only within a bin which many have few taxa? Do relative abundances play no role? 2) Further, I am confused at how these are calculated. I hope this gets clarified later in the text but some information here would be useful. Specifically, RC is based on comparing two different samples, but this is with a single bin within a single sample (is this correct?), how is this calculated if there are no other samples to compare it to? This needs to be clarified here and throughout.

Lines 142-151: So, based on this simulation, you made these the standard parameters for an iCAMP analyses? If so, when working with real data, what happens if these are not met, it might be difficult for some bins to have a minimum of 24 'species', what happens here? Does the tree force the smallest bin to be 24? My main point, I need more information as to how this ideal situation translates into biological reality full of noise and biases that may skew data? It might not always be fair to bin groups to conform to a pre-determined minimum as you would likely lose biological relevance.

Lines 192-214: This is really cool, but what is not addressed well is the idea that since we do not know the "true" nature of these communities, how do we know that the outputted iCAMP and QPEN results are good predictors when we don't know underlying distributions. I know this was optimized based on simulated data, but this point remains. Further, it appears based on these results that HoS was overrepresented using QPEN compared to iCAMP. It is a ~50% differential which is suppressing given that iCAMP was only 9.9% more accurate (but 120 more precise). So, is your interpretation that this is noise in QPEN? If so, this needs to be tested/explained. One easy way to demonstrate that iCAMP values are more 'correct' would be to iterate this analysis say 1000 times that the average values for HoS, etc. be used. This would do much to validate that iCAMP is a better and more 'true' way to examine these processes.

Lines 216-218: I find the word 'generally' here problematic. If they are generally considered to be truly stochastic thus, we can essentially ignore the noise, how can we really know this? What effect will this noise have on these estimates? Thus, I find this qualifier an issue.

Lines 218-223: Again, how can we be convinced that iCAMP is more 'correct'?

Line 226: Why is this obvious?

Lines 229-233: I think this is actually an argument against the importance of utilization of iCAMP over other methods – If all methods generally provide the same results, we are already equipped to understand these processes. Then the argument needs to be why iCAMP is better, and should not be – 'man, QPEN is really awful'. Right now, I haven't seen good rationale as to why iCAMP is a marked improvement and is needed. The bones are there, but it needs to be better explored.

Lines 239-241: This, in my opinion, is the best part of iCAMP. The ability to distill and infer sub-community level differences. Again, though, within a 'bin', is it reasonable to assume all members are members are being acted upon by the same processes? They could, for instance be acted upon by completely different processes due to the processes of niche differentiation or niche complementation which could easily allow closely related taxa (within a bin) to respond functionally differently. This must be addressed.

Lines 243-248: Given my bin membership concerns based on phylogenetic closeness, how accurate are these estimates of lineage-specific ecological processes? And, what does that actually tell us?

Discussion:

Line 284: I do not see why it is particularly 'more difficult' in microbial compared to macrobial ecology.

Lines 291-293: The wording of these sentences implies that most microbial ecologists are content with examining biodiversity patterns, a sentiment that I find unlikely. We all wish to better understand mechanisms, as all ecologists do, otherwise it is natural history. So let's not claim this.

Line 294: I do not like this sentence; it lacks logic for this extension of use. Either go all in or omit.

Lines 296-298: Exactly, an underlying assumption that I find not likely for all circumstances. In fact, there is a strong argument to be made that closely related organisms will outcompete each other (competitive exclusion) and thus in order for them to co-exist, niche differentiation must occur so then they would not have the same niche-space (line 298). I find this counter to your arguments and undercuts how iCAMP works. This needs to be addressed.

Line 299 and other: It occurs to me that this MS would benefit if the authors operationally define what is a high, medium, and low phylogenetic signal.

Lines 298-302: This may be true but does not address fully the concern of members of the same bin having the same 'niche space' which I find an unreliable assumption.

Lines 310-319: Given the majority (too much in my opinion) of your results are focused on how iCAMP outperforms QPEN, I feel you must do a better job justifying why these comparisons are so important. Why the disdain for QPEN? If you feel that QPEN is just bad at estimating these factors and you want to prevent folks from using so that they do not have spurious results and then to infer inappropriate patterns, that is one thing. But it just reads like you have QPEN for an unknown reason. I think it is important for the reader to know why. This is imperative to put your advancements in the proper light.

Lines 319-323: It is a bit anthropomorphizing to claim that these grasslands are not stressful. If there were not stressors, there would be no competition which of course does not happen.

Lines 339-344: This may be a bit speculative based on one grassland study.

Methods:

Lines 362-363: I am not sure what I think of the 'core taxa' label – core taxa in microbial ecology generally means one that is ubiquitous across the experimental framework. I worry using this same terminology to mean something else is unnecessarily confusing.

Lines 366-367 and Lines 376-377: This gets at an earlier point I made, if bins are merged, don't you lose the power to assume niche conservatism? It might be better to omit these bins from analysis than combine them.

Lines 381-394: I would like to learn more about this 'niche preference' approach. It does not seem to be that you would have enough information to identify a niche preference, and what does this even mean? This is a bit nebulous but it appears crucial to your approach so it must be treated as such and this requires justification for these assumptions.

Lines 431-434: I think this would be improved if you also did an iterative approach on the grassland to explore how many randomization steps are suggested for real data.

Lines 542-543: Is it reasonable to assume no dispersal limitations? Especially since the authors claim that iCAMP is suitable for animal-based systems as well?

Lines 618-619: This sentence is odd and needs to be rewritten for clarity.

Responses to Reviewers' comments

Manuscript number: NCOMMS-20-03950A

A. Reviewer #1 (Remarks to the Author):

This article presents an interesting approach to study phylogenetic diversity pattern of microbial communities. While I am not overly familiar with diversity pattern analysis in microbial studies, I think the approach is new and looks well rounded. The inclusion of a simulation study is remarkable as most newly devised diversity pattern methodologies dispense themselves from doing such a validation work. Furthermore while the analysis is undeniably complex, a functioning R-package was provided which suggests that the method will be easily used by other microbial ecologists

This being said, I have two major concerns, one on the methodology and one on the substance of the analysis. At the moment, I feel that the empirical case study do not convince me that iCAMP is a superior analysis to QPEN.

Reply: We appreciate the insightful comments which really helped us improve this study. Evidences demonstrating the advantages of iCAMP over QPEN are:

- iCAMP assessed the influence of assembly processes in each of the 658 phylogenetic groups, but QPEN is simply not able to do this. We believe that this is an obvious improvement. We emphasized this feature of iCAMP in the revision, e.g. “In contrast to QPEN and other approaches, ...”(main text Line 229-230 [tracking version Line 276]).
- For simulated data, iCAMP provides more precise estimation of relative importance of assembly processes (main text Line 155-175 [tracking version Line 193-219], Fig. 2, S9-S12).
- In the empirical case study, iCAMP provides more reasonable estimation than QPEN (main text Line 196-203 and 208-217 [tracking version Line 242-249 and 254-263]). Estimated by QPEN, the relative importance of stochastic processes is around 17-27%, and down to 0% in some years. Obviously, 0% stochasticity makes no sense for natural microbial communities which definitely have substantial proportion of random birth/death and non-deterministic passive migration of microbes. It is also conflict with widely reported important roles of stochastic processes across various ecosystems, even in desert and acidic soils (main text Line 201-203 [tracking version Line 247-249]). By iCAMP, the stochasticity is estimated as ~60%, which is much more reasonable than 0-20% and at similar level as the estimation by other published methods.

Methodological concerns:

(A1) While I may have missed it, I failed to find the test of the different null models combination (Figure S3). Phylogenetic beta diversity can be tested by both a within-bin randomization or a across-bin randomization, same for the taxonomic beta diversity. So four null models/indices

combinations should be tested.

Reply: Yes, we agree that the four combinations need to be tested. We revised Fig. S6 and supplementary text Line 63-68 [tracking version supp. Line 65-70].

(A2) When the main ecological process is assessed within bins, is it using the across-bin null model for the taxonomic beta-diversity (which I would deem inadequate for accessing the ecological process occurring within a given bin), or is it with the within-bin null model?

Reply: Thanks for raising this point. After the across-bin null model generates 1000 random datasets (randomized OTU tables), the null taxonomic beta-diversity is still calculated for each turnover of each bin, according to the OTU IDs of each bin. In this way, for each turnover of each bin, the observed beta-diversity value can be compared with the 1000 null beta-diversity values of the bin, no matter the null model is within-bin or across-bin. To avoid confusion, the method part Line 453-455 [tracking version Line 541-542] is added with a sentence “No matter whether the randomization is within or across bins, the beta diversity metrics are calculated in each bin as defined in Eq. 6-10.”

Based on the results from simulated communities (Fig. S6), the across-bin null model shows obviously higher accuracy, precision, and sensitivity for taxonomic beta-diversity to infer the effect of dispersal limitation or homogenizing dispersal although the within-bin null model is indeed better for phylogenetic beta-diversity to infer selection. We added the possible reasons for explanation in the supplementary text Line 69-74 [tracking version supp. Line 71-78]:

- For the point that the within-bin null model is better for phylogenetic diversity, “... mainly due to significant phylogenetic signal within bin rather than across bins, which is important for using β NRI to infer selection.”
- For the point that the across-bin null model is better for taxonomic diversity, “... is reasonable considering that the taxonomic null model analysis is used to infer neutral dispersal process, which is not species-specific but influences taxa across all bins in probability as long as under the same metacommunity.”

(A3) In the empirical experiment, it seems that warming mostly induces a change in bin abundances rather than changing the ecological process that occur within bins. As stated by the authors, the increasing of homogenizing selection seem to be mainly driven by an increasing relative abundance of Firmicutes that are under homogenizing selection regardless of the temperature treatment.

So, if the analysis suggests that the effect of warming plays a selective role between phylogenetic bins (i.e. by favouring Firmicutes over other clades), is it really relevant to analyse the effect of the warming treatment within bins?

Similarly, I am wondering what is the logic of not analysing the beta-diversities between the control and the treated communities, it seems that it would be an interesting way to access whether warming generates heterogeneous selection on microbial communities.

Reply: Thanks for the suggestions! The first question is whether it is relevant to analyze the warming effect within bins, when warming seems select some bins out of others. The fact is not that simple. For example, in the later 3 years, warming significantly increased the relative

abundance of 27 bins (Wilcoxon $P < 0.05$ and Cohen's $d > 0.5$, i.e. medium to large effect size), suggesting that the effect of warming played a selective role. However, in these apparent “selected” bins, 81% (22 bins) were not governed by selection under warming, and two bins governed by drift under control switched to be governed by selection under warming. An interesting example is Bin 3 in Alphaproteobacteria, the 3rd most abundant bin. Bin 3 showed higher relative abundance (23% higher) under warming than in control, which makes it look like “selected” by warming. However, based on iCAMP results, Bin 3 was governed more by drift (61% DR and 38% HoS) under warming and more by selection (61% HoS and 38% DR) under control.

To avoid the impression that selection can be inferred by simply higher abundances, we revised the main text Line 251-257 [tracking version Line 299-304] as below: “From Year 3 to Year 5, 27 bins showed significantly higher relative abundances under warming than control (Wilcoxon $p < 0.05$, Cohen's $d > 0.5$). However, 81% of them (22 bins) were mainly governed by drift, while two of them switched from governed by drift in controls to homogeneous selection under warming. These results demonstrated complex assembly mechanisms of different bins in response to warming, and possible misinterpretation could occur without bin-level mechanistic information.”

The second question is related to the beta-diversities between the control and the warmed communities. Our goal is to determine whether and how warming affect community assembly processes. Thus, it is more appropriate analyze the data within treatment or control rather than between treatment and controls. Also, if we include beta diversities between warming and controls, the between-condition results will make the description and interpretation much more complex, which is not our main purpose. To avoid distraction, we still insist in only discussing the mechanisms within each type of condition.

Nevertheless, considering between-treatment turnovers can be interesting in other studies, we have “between.group” option into the summary tools (functions `icamp.bins` and `icamp.boot`) in iCAMP package. We mentioned this in the Method part of the revision (main text Line 482-484 [tracking version Line 569-571]). In addition, with the same curiosity as the reviewer, we did the between-condition analysis for the empirical data. In all years, the relative importance of heterogeneous selection was $0.46 \pm 0.05\%$ in turnovers between warmed and control plots, which is slightly higher than that within warmed plots ($0.40 \pm 0.06\%$). The homogeneous selection is still very important ($38 \pm 1\%$) in governing the turnover between warmed and control plots. It is reasonable because the communities in warmed and control plots share some important selective pressures other than (even stronger than) the mild 2-degree warming (e.g. daily and seasonal temperature change, seasonal and yearly precipitation change, etc.).

(A4) Form concern: I feel that the analysis is too layered to be easily grasped by the reader. I understand that the authors want to present a thorough analysis but even an attentive reader like me could feel lost. It is aggravated by the introduction-results-discussion-methods layout of Nature Communications that is not ideal to handle such a methodological paper. I would recommend the author to trim the article and put (even) more analytical elements in the

supplementary materials, especially all the ones are not essential to understand the methodology and grasp its value (good candidates would be the comparison of the algorithms hacking the trees to produce phylogenetic bins and the comparison of various diversity indices).

Reply: We fully agree it is better to be more concise. We simplified the “optimization” part (main text Line 135-138 [tracking version Line 161-175]) and left detailed description of this part in supplementary text.

(A5) I recommend limiting the use of acronyms. While it is perfectly ok for algorithms and diversity index names, when it is about ecological processes (HoS etc), I find them unpleasant to read.

Reply: We agree and changed to use full names for ecological processes.

(A6) Various remarks:

*l. 38: the most diverse group*** - 'Microorganisms' are paraphyletic*

Reply: Revised.

(A7) l.40: 'are critical to our society' - they are critical to ecosystems in general really.

Reply: Revised to “... are critical to ecosystem functioning and service.”

(A8) l.109 - 110: how were determined those thresholds?

Reply: The thresholds are commonly used, i.e. 1.96 (or 2 in some references) for betaNRI or betaNTI (e.g. Fine & Kembel 2011; Stegen et al 2012; Gastauer et al 2015; Liu et al 2017) and 0.95 for RC (e.g. Chase et al 2011; Stegen et al 2013). Statistically, these thresholds are corresponding to the probability of 0.95 ($P=0.05$), the most widely used threshold of significance. We cited some of the references in the revision (Line 104 and 107 [tracking version Line 129 and 133]).

References:

- [1] Fine PVA, Kembel SW. 2011. Phylogenetic community structure and phylogenetic turnover across space and edaphic gradients in western Amazonian tree communities. *Ecography* 34:552-565. doi:10.1111/j.1600-0587.2010.06548.x.
- [2] Stegen JC, Lin X, Konopka AE, Fredrickson JK. 2012. Stochastic and deterministic assembly processes in subsurface microbial communities. *ISME Journal* 6:1653–1664.
- [3] Gastauer M, Saporetti-Junior AW, Magnago LFS, Cavender-Bares J, Meira-Neto JAA. 2015. The hypothesis of sympatric speciation as the dominant generator of endemism in a global hotspot of biodiversity. *Ecology and evolution* 5:5272-5283. doi:10.1002/ece3.1761.
- [4] Liu C, Yao M, Stegen JC, Rui J, Li J, Li X. 2017. Long-term nitrogen addition affects the phylogenetic turnover of soil microbial community responding to moisture pulse. *Scientific Reports* 7:17492. doi:10.1038/s41598-017-17736-w.

[5] Chase JM, Kraft NJB, Smith KG, Vellend M, Inouye BD. 2011. Using null models to disentangle variation in community dissimilarity from variation in alpha-diversity. *Ecosphere* 2:1-11. doi:10.1890/es10-00117.1.

[6] Stegen JC, Lin X, Fredrickson JK, Chen X, Kennedy DW, Murray CJ, Rockhold ML, Konopka A. 2013. Quantifying community assembly processes and identifying features that impose them. *The Isme Journal* 7:2069. doi:10.1038/ismej.2013.93.

(A9) L. 210: can you suggest why the

Reply: This comment appears incomplete. The reason for inconsistent results from QPEN could be that the analysis is at the whole community level. However, the actions of various ecological processes are more likely at finer levels such as populations, lineages and etc.

(A10) l. 254: Do you have an interpretation regarding the increase of Firmicutes abundances in the control plot for the first three years?

Reply: It is not precise to say the Firmicutes Bin 1 “increase” in the control in all the first three years. From Year 1 to Year 2, Bin 1 did not significantly change (the mean value even slightly “decreased”) in controls, but significantly increased in warmed plot. From Year 2 to Year 3, corresponding to significant drought in the sampling month of Year 3, Bin 1 increased in both control and warmed plots, but the mean relative abundance in controls had become slightly lower than in warmed plots. Warming has made difference every year, including the first three years.

To be clearer we revised the sentence to “After Year 3, which had severe drought, Firmicutes were significantly more abundant under warming than control”.

(A11) Figure 3 : pretty but incomprehensible. Barplots are misleading, rather see it on a 0-1 scale. I am not convinced that overlaying effect size and barplot helps the interpretation.

Reply: Revised the scale to 0-100% for all bar plots, and removed the lines of effect size.

(A12) Figure 4 : Too many redundant colours, maybe a way to simplify the figure would be to class more bins into the 'other' category.

Reply: Revised. Reduced colors.

(A13) Fig S14: I would suggest to test the model HoS ~ Drought + Plant cover to distinguish the two effects. I am suspecting that drought and plant cover are related. Partial mantel test could be used for instance.

Reply: Partial Mantel test was performed for the factors showing significant correlation with homogeneous selection or drift. Interestingly, the plant biomass under warming and the variation of C4 plant biomass in controls showed very small change in correlation with homogeneous selection or drift when drought or any other variable are partitioned out. The results are presented in Table S3 and main text Line 272-274 [tracking version Line 322-325].

(A14) Figure 5 : Similarly a multi factor model should maybe be preferred. All these factors are likely to be somewhat correlated. Mantel test extensions like multiple regressions on distance matrices could be used <https://link.springer.com/article/10.1007/s11258-006-9126-3>.

Reply: We refined our MRM (multiple regressions on distance matrix) analyses, presented the results in Fig. 5b, 5c, S16b and S16c, main text Line 274-285 and 290-293 [tracking version Line 325-335 and 353-356]. Previous description of correlations in supplementary text becomes redundant, and thus was deleted.

B. Reviewer #2 (Remarks to the Author):

This MS is on the development of a new way to examine community assembly processes for microbial community data. It is certainly a worthwhile field to study as the authors correctly point out that these processes which underlay almost all community dynamics are, so far, not accounted for in any meaningful way. I applaud the amount of work and detail that went into this MS. Further, besides a few places (see below), it is a very well-written MS.

Despite the potential great advance this work may yield, there are a few major concerns I had. Primarily, these should be able to be addressed with better justification and some simulation work. Line by line notes are presented below.

Reply: Thanks for all the insightful comments.

(B1) The major concerns I have are more that I take some issues with the assumptions made about how these taxa would behave that underlay how these assembly processes are derived. Namely, it is fair to assume that all members within a bin have the same niche-space and functional roles, thus is it fair to assume that all members of the bin would have the same partition of the assembly processes. As written, I would say no, and given that this is a MAJOR baseline assumption, this casts suspicion on how the rest of the data that results can be trusted. I recommend conducting a simulation exercise that will demonstrate that this underlying assumption (even if false) will not impact obtained results much.

Reply: We fully agree with the reviewer's concerns. When QPEN identifies only one governing process for each turnover of the entire community, we raised a very similar question, "are all members within a community have the same partition of the processes?". We believe not. Thus, we developed iCAMP to mitigate this issue. The members of a bin are surely possible to have different governing processes. Thus, following the reviewer's suggestion, we did more simulation. We simulated bins with different actual size and phylogenetic distance thresholds, then when we still use the previous threshold (0.2) and size limitation (24), many estimated bins have members governed by different processes. Although the performance decreased when more individual bins governed by multiple processes, iCAMP showed promising robustness and remained relatively high performance (quantitative accuracy > 0.98, precision > 0.68), even if 68% bins (up to 100% in some situations) were governed by mixed effects of different processes. We presented the results in Fig. S10, main text Line 162-163, and supplementary text B Line 128-143 [tracking version supp. Line 184-199].

(B2) Also, is it fair to combine bins to meet the minimum size, even if these were originally binned separately (I am not convinced)?

Reply: Yes, combining bins has drawbacks, e.g. it may mix more members under different assembly mechanisms to the same bin. But, if not combined, quite a few original bins usually have too few species (even less than 3), leading to too low statistical power and making null model analysis impossible or highly unreliable. Thus, although it is a compromise, combining bins is necessary in many cases. The binning algorithm ensures each small bin is combined with the nearest relative bin. And the bin size limitation (minimal bin size) is determined by targeting optimum within-bin phylogenetic signal which is crucial to infer selection. In the revision, (i) we provided the functions "dniche" and "ps.bin" in iCAMP package and mentioned an alternative way to determine binning settings based on within-bin phylogenetic signal (supplementary text

Line 36-42, main text Line 418-420 [tracking version Line 505-507]). (ii) we also simulated datasets with different actual bin sizes (down to 3 species) and performed iCAMP with fixed bin size setting (at least 24 species per bin). The results showed the robustness of our method, as described above (the reply to the question B1). (iii) Considering it may be applicable to just omit the small bins (not the case in our manuscript), we added an option to omit small bins in the iCAMP package, and the tool will also return the list of omitted taxa to avoid removal of important taxa. Another way to alleviate this problem is to perform much deeper sequencing which may be able to discover enough species in small bins. But this is beyond the scope of the current study.

(B3) Further, I feel the authors use very strong language, almost aggressive language, when they compare their results against QPEN, but provide no justification for such seemingly denegation. The authors may have their reasons, but the reader is not privy to this.

Reply: Thanks for pointing this out. We have had been very careful about language used, but apparently, it is still not enough. In one version, we had tried to delete any direct comparison between iCAMP and QPEN, but this did not read well scientifically. Thus, we put it back and also comparison with other methods by not simply singling out QPEN (See Fig. 2a-c and Fig. 3).

Now, we checked through the text and tried our best to revise the language, with some examples as below.

[Examples]

Main text Line 166-168 [tracking version Line 206-211]: revise “QPEN had lower performance (quantitative precision < 0 , qualitative precision < 0.13 , sensitivity < 0.04 , Fig. S10) in estimating HoS and HeS even under high phylogenetic signal” to “... indicating considerable improvement from QPEN, particularly in estimating homogeneous and heterogeneous selection”.

Main text Line 170 [tracking version Line 212-213]: revise “despite that it was considerably higher than QPEN” to “albeit still higher than QPEN”.

Main text Line 323-324 [tracking version Line 387-388] revise “QPEN did not perform well in inferring selection with the simulated communities” to “QPEN did not always perform well in inferring selection (e.g. in our simulated communities)”.

We agree that our statements from comparison should have “justification” and “reasons”. We provided data and figures to justify that we can say iCAMP has better performance than QPEN in the simulated communities (e.g. Line 147-175, Fig. 2, S9-S12) and discussed the phylogenetic signal issue (Line 323 to 328). In this revision, we chose to delete a previous paragraph about the comparison between iCAMP and QPEN on stochasticity estimation in the empirical data, to avoid unnecessary emphasis.

(B4) Finally, given line 88, can iCAMP be modified to run with gene data (each gene is an OTU)? It is implied that this is needed but it does not appear that iCAMP can.

Reply: It is possible as long as there is some phylogenetic conservatism for the different sequences of a functional gene in different species. If integrating different gene families, the method can be more informative. We are developing new methods in this direction. We did not

add any discussion but just mentioned “functional traits (genes)” in the last sentence, considering this is beyond the focus of this manuscript.

(B5) Abstract: Line 31: Change “on Bacillales” to ‘on the Bacillales’

Reply: Revised.

(B6) Line 31 -32: This sentence is awkward. What is strengthened exactly. The ‘selection’ or the impact of selection on community assembly. This just needs to be a bit more precise.

Reply: Thanks! This sentence was changed to “Interestingly, warming decreased “drift” over time, and enhanced homogeneous selection, which is largely upon the Bacillales. Also, homogeneous selection had higher correlations with drought and plant productivity under warming than control.”.

(B7) Line 33: Strengthen this closing sentence.

Reply: Thanks. This sentence was revised to “The general framework described here provides an effective and robust tool to quantify community assembly processes in microbial ecology, and it should also be useful for plant and animal ecology.”

(B8) Introduction: Line 42: What mechanisms are referred to here? This is a very open statement and the challenge of understanding various mechanisms is not constant.

Reply: Revised to “the mechanisms controlling biodiversity and community composition”.

(B9) Liens 42-43: Is NGS development still ‘recent’? lines exactly like this one have been used for the past decade.

Reply: Deleted “recent”. Slight change was made. Here, we would like to remind that there are quite a few developments in high-throughput metagenomic technologies in recent years and now, e.g. Illumina NovaSeq (2017-2018), PacBio Sequel II system (2019), GeoChip 5.0 (published 2019, we are developing GeoChip 6.0).

(B10) Lines 44-45: Why?

Reply: Thanks! This sentence was revised to “ However, analyzing such massive data to address fundamental ecological questions such as community assembly mechanisms is challenging due to various issues associated with detection specificity, sensitivity, quantification, reproducibility, and taxonomic resolution (Zhou et al. 2015).”

(B11) Liens 50-51: I take issue with the presumption that these processes are stochastic. Stochasticity plays a large role of course, but there are many other factors that are not random and it is the combined action that drives the rates.

Reply: Thanks! We revised to “largely controlled by...” and also added a little bit more details about deterministic factors: “Niche-based theory asserts that deterministic processes, including environmental filtering (e.g., pH, temperature, moisture, and salinity) and various biological

interactions (e.g., competition, facilitation, mutualisms, predation, and tradeoffs), largely control the patterns of species composition, abundance and distributions. In contrast, neutral theory assumes that all species are ecologically equivalent, and species dynamics is largely controlled by stochastic processes of birth/death, speciation/extinction, and immigration”.

(B12) Line 51: Awkward phrase

Reply: Revised to “After intensive decade’s debates, ...”

(B13) Line 56: “Umbrella” seems an odd choice of word here.

Reply: Revised to “To unify niche and neutral perspectives on governing community structure, ...”.

(B14) Lines 62-64: Why?

Reply: Thanks! Slight change was made of this sentence to “Due to the lack of quantitative approaches, most analyses with respect to the relative importance of the four processes across different types of natural communities are qualitative and subjective, and replete with great uncertainty”.

(B15) Lines 64-80: I feel like there is too much detail here, do we need a full breakdown of operations definitions here? Particularly given that you latter say that framework is limited, thus justifying your new framework.

Reply: Thanks for pointing this out. The detailed explanation for different terms were moved to Method section.

(B16) Line 82-84: While there is no doubt that this is an advance, I do not find it fair to claim that this allowed quantification of assembly processes in microbes for the first time. Many examples exist there careful experimentation has allowed such quantification (even as early as the 80s). However, not with this statistical power, but please be careful to not overpromise novelty.

Reply: Thanks! This sentence was revised to “This statistical approach represents a significant advance in microbial ecology that enables microbial ecologists to obtain quantitative information on community assembly processes.”

(B17) Lines 85-88: Is there not an argument to be made that if communities change, then lower level of organizations will change too, such as populations and gene complements? Of course I am not arguing that only communities matter, but to illustrate the importance for careful selection of words.

Reply: Thanks! The sentence was revised to “However, a major limitation is that various ecological processes are estimated based on the pairwise turnovers of the whole communities. This may not be appropriate because it is well known that the actions of various ecological processes (e.g. natural selection) are typically on the finer biological levels, such as genes, genotypes, cells, individuals, populations, and lineages.”

(B18) Line 91-92: Awkward phrasing.

Reply: Revised to “Also, various groups of organisms differ greatly in their responses to environmental changes.”

(B19) One thing that is missing is the direct statement of what iCAMP compares. There is some confusing language that make it uncertain if this compares all data from multiple samples or on a sample by sample basis. Give beta-NRI and RC, if must be across samples, but this should be explicitly stated.

Reply: Added “which relies on the turnovers of individual bins across communities (samples).” to the sentence defining iCAMP.

(B20) Results: Lines 106-107: Is it fair to assume that each bin (members of closely related organisms) should all behave similarly with respect to ecological processes. It is quite easy to contrive an example where even strain level variation (single terminal node depending on binning) would differ in their relative impacts to drift, selection etc. I would require some deep discussion and simulations to believe that this could not have an effect of process outcomes. This ignores competitive exclusion. Is nice conservation by closely related organisms the only option? I would say no.

Reply: We agree and appreciate the suggestions. We did more simulations of possibly mixed governing processes within individual bins. Although the performance decreased when more individual bins governed by multiple processes, iCAMP showed promising robustness and remained relatively high performance (quantitative accuracy > 0.98, precision > 0.68), even if 68% bins (up to 100% in some situations) were governed by mixed effects of different processes. We presented the results in Fig. S10, main text Line 162-163, and supplementary text B Line 128-143 [tracking version supp. Line 184-199].

In addition, the effect of competition is also added to our new simulations. Abiotic filtering and biotic competition are both simulated and given different ratios as different simulated situations. The results were presented in Fig. S11, main text Line 162-163, and supplementary text Line 145-158 [tracking version supp. Line 201-214].

(B21) Lines 111-114: Traditionally, Raup-Crick measures are binary in nature and are the probabilistic inferring if one selection of fossils is a random subset of another selection. 1) Is a binary matrix approach powerful enough be used in fine scale parsing of processes, especially if this is only within a bin which many have few taxa? Do relative abundances play no role? 2) Further, I am confused at how these are calculated. I hope this gets clarified later in the text but some information here would be useful. Specifically, RC is based on comparing two different samples, but this is with a single bin within a single sample (is this correct?), how this is this calculated if there are no other samples to compare it to? This needs to be clarified here and throughout.

Reply: In our manuscript, RC is not the original binary Raup-Crick index, but the modified Raup-Crick based on Bray-Curtis index, an abundance-weighted beta-diversity index, which was reported by Dr. Stegen in 2013 (Stegen et al 2013). Each RC value of a bin is calculated by the

comparison of taxa in the bin between two samples, which is explicit in the equation of RC (Eq.7, 8, 10), as well as other metrics (Eq. 1 to 10).

Reference:

- [1] Stegen, J. C. et al. Quantifying community assembly processes and identifying features that impose them. *The Isme Journal* 7, 2069, doi:10.1038/ismej.2013.93 (2013).

(B22) Lines 142-151: So, based on this simulation, you made these the standard parameters for an iCAMP analyses? If so, when working with real data, what happened if these are not met, it might be difficult for some bins to have a minimum of 24 'species', what happens here? Does the tree force the smallest bin to be 24? My main point, I need more information as to how this ideal situation translates into biological reality full of noise and biases that may skew data? It might not always be fair to bin groups to conform to a pre-determined minimum as you would likely lose biological relevance.

Reply: We agree. The parameters should not be pre-determined. For real data, the parameters had better be determined by targeting optimum within-bin phylogenetic signal. We did so for the empirical data in this manuscript. We realize this is important, and added functions “dniche” and “ps.bin” into our package to help determine the setting. We revised the description in method (main text Line 413-420 [tracking version Line 500-507]) and detailed in supplementary text Line 35 to 41, to address this issue.

[Method Line 413-420]

An optimized n_{min} should give the highest number of bins with significant phylogenetic signal and relatively high average correlation coefficient (average R) within bins. Accordingly, the optimized n_{min} is identified as 24 for simulated datasets and 12 for the empirical data. ... Otherwise, an alternative way should be considered, i.e. choose the n_{min} value that makes the estimated relative importance of stochastic processes close to the stochasticity from referable approaches (e.g. pNST).

[Supplementary text Line 35 to 41]

In empirical studies, if key environmental variables are available, the function “dniche” and “ps.bin” in iCAMP package can be used to calculate within-bin phylogenetic signal to determine n_{min} . If key environmental variables are unknown or unmeasured, n_{min} can be determined in an indirect way: (i) estimating stochasticity level with other approaches, such as phylogenetic normalized stochasticity ratio (pNST), which had better quantitative performance in the simulated communities (Fig. 2a-c); (ii) then, testing different n_{min} values and choosing the one with estimated stochasticity similar to pNST and/or other approaches.

(B23) Lines 192-214: This is really cool, but what is not addressed well is the idea that since we do not know the “true” nature of these communities, how do we know that the outputted iCAMP and QPEN results are good predictors when we don't know underlying distributions. I know this was optimized based on simulated data, but this point remains. Further, it appears based on these results that HoS was overrepresented using QPEN compared to iCAMP. It is a ~50% differential which is suppressing give that iCAMP was only 9.9% more accurate (but 120 more

precise). So, is your interpretation that this is noise in QPEN? If so, this needs to be tested/explained. One easily way to demonstrate that iCAMP values are more ‘correct’ would be to iterate this analysis say 1000 times that the average values for HoS, etc. be used. This would do much to validate that iCAMP is a better and more ‘true’ way to examine these processes.

Reply: We agree that we do not know the “true” relative importance of stochastic processes in an empirical study. But we can tell which is likely not reasonable. While iCAMP and other published methods implicated that there is obvious stochasticity, QPEN estimated 0% stochasticity in several years, even in control plots. Considering substantial stochasticity in microbial assembly has been reported across various ecosystems (e.g. Ofiteru et al 2010; Zhou et al 2013, 2014; Evans et al 2017; Wu et al 2019), even in desert (Caruso et al 2011) and acidic soil (Tripathi et al 2018), we believe that iCAMP’s results are more reasonable. A sentence was added to main text Line 201-203 [tracking version Line 247-249].

References:

- [1] Ofiteru, I. D. et al. Combined niche and neutral effects in a microbial wastewater treatment community. *Proceedings of the National Academy of Sciences of the United States of America* 107, 15345-15350, doi:10.1073/pnas.1000604107 (2010).
- [2] Zhou, J. et al. Stochastic assembly leads to alternative communities with distinct functions in a bioreactor microbial community. *mBio* 4, e00584-00512, doi:10.1128/mBio.00584-12 (2013).
- [3] Zhou, J. et al. Stochasticity, succession, and environmental perturbations in a fluidic ecosystem. *Proceedings of the National Academy of Sciences of the United States of America* 111, E836-E845, doi:10.1073/pnas.1324044111 (2014).
- [4] Evans, S., Martiny, J. B. H. & Allison, S. D. Effects of dispersal and selection on stochastic assembly in microbial communities. *ISME J* 11, 176-185, doi:10.1038/ismej.2016.96 (2017).
- [5] Wu, L. et al. Global diversity and biogeography of bacterial communities in wastewater treatment plants. *Nature Microbiology* 4, 1183–1195, doi:10.1038/s41564-019-0426-5 (2019).
- [6] Caruso, T. et al. Stochastic and deterministic processes interact in the assembly of desert microbial communities on a global scale. *ISME J* 5, 1406-1413, doi:10.1038/ismej.2011.21 (2011).
- [7] Tripathi, B. M. et al. Soil pH mediates the balance between stochastic and deterministic assembly of bacteria. *ISME J* 12, 1072-1083, doi:10.1038/s41396-018-0082-4 (2018).

In addition, although the overall quantitative accuracy of iCAMP is only 9.9% higher than QPEN when counting the average in all processes and all simulated situations/scenarios, for homogeneous selection (HoS) only (Fig. S12), the quantitative accuracy of iCAMP is 165% to 676% higher than QPEN. For stochasticity estimation in simulated datasets, the quantitative accuracy of iCAMP is 296% to 864% higher accuracy than QPEN (Fig. 2a-c). Thus, the large difference between iCAMP and QPEN results in empirical data is not surprising.

Nevertheless, to thoroughly address the reviewer's concern, we have further tested the reproducibility (noise from null model analysis) in QPEN and iCAMP. For iCAMP, we previously tested different randomization times (from 25 to 5000) in null models for simulated data (Fig. S7). For each option, we repeated 12 times and found small deviation from null expectation (<2.8%, i.e. high reproducibility), with IQR values from 0.00% to 1.82% and Range values (max-min) from 0.07% to 2.8% for different processes, when using 1000 or more randomization times. We did similar test for QPEN, which also showed small deviation (<6.7%) with IQR values equal to 0% and Range values from 2.8% to 6.7%, when using 1000 or higher randomization times. For the empirical data, we also repeated iCAMP and QPEN 3 times, with the commonly used 1000-time randomization in null models. iCAMP can estimate relative importance of each process in the 780 pairwise comparisons among the 40 samples. In the 780 results of each process, the standard deviation among these 3 tests ranges from 0 to 6.8%, with IQR values from 0.03% to 1.02% for different processes, indicating sufficient reproducibility. QPEN can identify one dominating process for each of the 780 pairwise comparisons. In the 780 estimates, only 15 (1.9%) are different among these 3 tests. These results were added to supplementary text Line 85 to 97 [tracking version supp. Line 92-104] in the revision. Altogether, the “overrepresented HoS” by QPEN appears to be not due to the noise of the methods.

(B24) Lines 216-218: I find the word ‘generally’ here problematic. If they are generally considered to be truly stochastic thus, we can essentially ignore the noise, how can we really know this? What effect will this noise have on these estimates? Thus, I find this qualifier an issue.

Reply: We revised this sentence to be clearer. The null model analysis means to distinguish non-random patterns from random ones. Thus, in principle, the “selection” effects identified by the null model can be attributed to selection and/or deterministic parts of other processes (e.g. non-neutral dispersal by active propulsion). In contrast, the effects of “dispersal” and “drift” identified from null models are probabilistic thus largely stochastic (Zhou & Ning 2017). So, we revised the sentence to “Based on the principle of the null models employed by iCAMP and QPEN, the fractions of dispersal limitation, homogenizing dispersal, and drift are largely considered stochastic⁵. Thus ...” (main text Line 205 to 207 [tracking version Line 251-253]).

Reference:

[1] Zhou, J. & Ning, D. Stochastic community assembly: Does it matter in microbial ecology? *Microbiology and Molecular Biology Reviews* 81, doi:10.1128/membr.00002-17 (2017).

We are not quite sure what the term “noise” means here. The “noise” can be from technical noise from null model, and noise from randomness among biological replicates. The noise from null model should be low when randomization times are high enough, as described above in the reply to the question B23. Bootstrapping analysis can help to reveal randomness among biological replicates. We presented the accuracy, precision, and their variation (error bars in Fig. 2a-c) of different stochasticity indexes applied to simulated data (Fig. 2a-c), and the index from iCAMP shows advancement.

(B25) Lines 218-223: Again, how can we be convinced that iCAMP is more ‘correct’

Reply: No, we did not and should not say it is more correct. But we could say iCAMP results are more reasonable, as described in the reply to the question B23. We added “...This appears not reasonable, considering important roles of stochastic processes have been widely reported across various ecosystems, including desert and acidic soils.” in main text Line 201 to 203 [tracking version Line 247-249].

(B26) Line 226: Why is this obvious?

Reply: Revised to “It appears that VPA, NST, and NP showed more consistent results with iCAMP than QPEN.” We said “obviously”, because, VPA/NST/NP results range from 49% to 79% on average, while iCAMP results are around 62% (differences are 6% to 17%) and QPEN results are lower than 27% (differences are 25% to 62%). 6-17% is obviously less than 25-62%.

(B27) Lines 229-233: I think this is actually an argument against the importance if utilization of iCAMP over other methods – If all methods generally provide the same results, we are already equipped to understand these processes. Then the argument needs to be why iCAMP is better, and should not be – ‘man, QPEN is really awful’. Right now, I haven’t seen good rationale as to why iCAMP is a marked improvement and is needed. The bones are there, but it needs to be better explored.

Reply: This paragraph is not about to argue iCAMP is better than all other approaches, but more intends to discuss the scientific question, whether stochastic processes played some roles and how warming affected the stochasticity, based on the results from different approaches. We do not know the “true” stochasticity in the real microbial communities, and every approach has its own advantages and defects. We have to rely on multi-thread evidences to increase our confidence, to get some sense on the reasonable level of stochasticity. As a result, inevitably, it reads like “one (or several) is awful than many others” rather than “one is better than all others”.

As to quantify selection/dispersal/drift, all other approaches are not comparable with iCAMP and QPEN. VPA, NST, and NP are simply not able to do so, as we have mentioned in Line 155 [tracking version Line 193] (“only QPEN and iCAMP can ...”). Thus, we mean to focus on comparison between iCAMP and QPEN. Based on the experimental results, it appears that iCAMP gave more reasonable results.

(B28) Lines 239-241: This, in my opinion, is the best part of iCAMP. The ability to distill and infer sub-community level differences. Again, though, within a ‘bin’, is it reasonable to assume all members are members are being acted upon by the same processes? They could, for instance be acted upon by completely different processes due to the processes of niche differentiation or niche complementation which could easily allow closely related taxa (within a bin) to respond functionally differently. This must be addressed.

Reply: We greatly appreciate the reviewer points out this important issue. Yes, members in a bin can be governed by different processes. In the revision, we tested the robustness of iCAMP to different processes in a bin with the new simulations. Although the performance decreased when more individual bins governed by multiple processes, iCAMP showed promising robustness and remained relatively high performance (quantitative accuracy > 0.98, precision > 0.68), even if

68% bins (up to 100% in some situations) were governed by mixed effects of different processes. We presented the results in Fig. S10, main text Line 162-163, and supplementary text B Line 128-143 [tracking version supp. Line 184-199].

(B29) Lines 243-248: Given my bin membership concerns based on phylogenetic closeness, how accurate are these estimates of lineage-specific ecological processes? And, what does that actually tell us?

Reply: In simulated data, we added estimation of quantitative performance at bin level (Fig. S10g-i) when testing different situations where each bin can be governed by multiple processes. iCAMP showed robustness at both community (Fig. S10d-f) and bin levels (Fig. S10g-i, supplementary text Line 138 to 143 [tracking version supp. Line 194-199]). When bins can actually be governed by multiple processes, the process identified for a bin means to represent the dominating process in the bin. For empirical data, it is hard to prove whether the results are accurate or not, because we do not know the “truth”.

(B30) Discussion:

Line 284: I do not see why it is particularly ‘more difficult’ in microbial compared to macrobial ecology.

Reply: In microbial compared to macrobial ecology, it is more difficult to accurately count individuals in a sample, e.g. current sequencing technology still has problems in quantification (Zhou et al 2011, 2013, 2015), given the sequencing usually only covers 10^4 to 10^5 individuals while there are 10^8 - 10^9 individuals per gram soil; more difficult to measure traits or niche preference of each specie (it is commonly believed 99.9% species are unculturable); more difficult to measure or track dispersal (migration) (Albright et al 2018; Nemergut et al 2013); more difficult to apply statistical approaches given the large number of species and individuals, particularly in datasets with many samples across large scales; even the definition of a species is still under debate in microbial ecology (Konstantinidis et al 2006; Rossello-Mora & Amann 2015); and so on. In the revision, we mentioned “...– mainly due to the huge diversity of microorganisms and the difficulties in their detection and quantification”, and cited some references for this point, to keep concise.

References:

- [1] Zhou, J. et al. Reproducibility and quantitation of amplicon sequencing-based detection. *ISME J* 5, 1303-1313, doi:10.1038/ismej.2011.11 (2011).
- [2] Zhou, J. et al. Random sampling process leads to overestimation of β -diversity of microbial communities. *mBio* 4, e00324-00313, doi:10.1128/mBio.00324-13 (2013).
- [3] Zhou, J. et al. High-throughput metagenomic technologies for complex microbial community analysis: Open and closed formats. *mBio* 6, e02288-02214, doi:10.1128/mBio.02288-14 (2015).
- [4] Albright, M. B. N. & Martiny, J. B. H. Dispersal alters bacterial diversity and composition in a natural community. *The ISME Journal* 12, 296-299, doi:10.1038/ismej.2017.161 (2018).

- [5] Nemergut, D. R. et al. Patterns and processes of microbial community assembly. *Microbiology and Molecular Biology Reviews* 77, 342-356, doi:10.1128/mnbr.00051-12 (2013).
- [6] Konstantinidis, K. T., Ramette, A. & Tiedje, J. M. The bacterial species definition in the genomic era. *Philos Trans R Soc Lond B Biol Sci* 361, 1929-1940, doi:10.1098/rstb.2006.1920 (2006).
- [7] Rossello-Mora, R. & Amann, R. Past and future species definitions for Bacteria and Archaea. *Systematic and Applied Microbiology* 38, 209-216, doi:10.1016/j.syapm.2015.02.001 (2015).

(B31) Lines 291-293: The wording of this sentences implies that most microbial ecologists are content with examining biodiversity patters, a sentiment that I find unlikely. We all wish to better understand mechanisms, as all ecologist do, otherwise it is natural history. So let's not claim this.

Reply: Thanks for reminding the misleading language. We revised the sentence to “The developed framework would provide an effective and robust tool to quantify community assembly processes in microbial ecology towards mechanistic understanding of community diversity and succession.” Main text Line 311-313 [tracking version Line 374-376]

(B32) Line 294: I do not like this sentence; it lacks logic for this extension of use. Either go all in or omit.

Reply: Yes, we should show the logic. We revised the sentence as below (Line 313-315 [tracking version supp. Line 376-379]).

“Given that our framework and simulation are general to high diversity communities rather than specific to microbes, it should also be applicable to plant and animal communities with high diversity.”

(B33) Lines 296-298: Exactly, an underlying assumption that I find not likely for all circumstances. In fact, there is a strong argument to be made that closely related organisms will outcompete each other (competitive exclusion) and thus in order for them to co-exist, niche differentiation must occur so then they would not have the same niche-space (line 298). I find this counter to your arguments and undercuts how iCAMP works. This needs to be addressed.

Reply: Yes, the theory is very reasonable and can be true in many cases. However, the fact is “significant phylogenetic signals were frequently found within a short phylogenetic distance (e.g. Wang et al 2013; Stegen et al 2012; Tripathi et al 2018)”, based on which Stegen et al. developed QPEN and we further developed iCAMP from QPEN.

Nevertheless, the reviewer did raise the right question, leading to reasonable doubt on the validity of using phylogenetic diversity metrics to infer selection (i.e. environment filtering, competition, and other biotic interactions). We did new simulation to include different ratios of

competition. iCAMP showed robustness, however, the decrease of iCAMP performance is obvious when competition predominates (Supplementary Text B Line 152-155 [tracking version supp. Line 208-211]).

However, the situation that competition disrupts the niche conservatism within short phylogenetic distance should not happen quite often, otherwise the significant phylogenetic signal within short phylogenetic distance would not be observed so widely. We added warning to the decrease of iCAMP performance when competition is strong (as below).

Discussion, main text Line 328 to 333 [tracking version Line 399-414]

However, if competition predominates, closely related organisms may lead to strong competitive exclusion, which could disrupt phylogenetic conservatism, and thus significantly decrease the precision of iCAMP as showed in our simulation. This situation is possible and deserves attention. But given that the significant phylogenetic signal within short phylogenetic distances is widely observed, the disruption may not happen often under natural settings as theoretically predicted.

References:

- [1] Wang, J. J. et al. Phylogenetic beta diversity in bacterial assemblages across ecosystems: deterministic versus stochastic processes. *ISME J* 7, 1310-1321, doi:10.1038/ismej.2013.30 (2013).
- [2] Stegen, J. C., Lin, X., Konopka, A. E. & Fredrickson, J. K. Stochastic and deterministic assembly processes in subsurface microbial communities. *ISME J* 6, 1653–1664 (2012).
- [3] Tripathi, B. M. et al. Soil pH mediates the balance between stochastic and deterministic assembly of bacteria. *ISME J* 12, 1072-1083, doi:10.1038/s41396-018-0082-4 (2018).

(B34) Line 299 and other: It occurs to me that this MS would benefit if the authors operationally define what is a high, medium, and low phylogenetic signal.

Reply: In this sentence, we used “(i.e. close to or lower than Brownian Motion expectation)” to define “medium or low” phylogenetic signal mentioned here. For the simulated scenarios, we defined high, medium, and low signal with Blomberg’s K (main text Line 127 [tracking version Line 153]), i.e. low (Blomberg’s K = 0.15), medium (K = 0.9), and high (K = 5.5). In the method (main text Line 527, 534, 536 [tracking version Line 614, 621, 623]), we explained why we can define so: “... Blomberg’s K value is as low as 0.15, close to the mean K value of 91 continuous prokaryotic traits, ... K value is 0.9, close to the mean phylogenetic signal level of 899 prokaryotic binary traits.... K value is as high as 5.5, close to the highest phylogenetic signal of prokaryotic traits to date.”

(B35) Lines 298-302: This may be true but does not address fully the concern of members of the same bin having the same ‘niche space’ which I find an unreliable assumption.

Reply: We agree that it is problematic to assume “the members of the same bin have the same niche space”. iCAMP does not hold such assumption. Instead, iCAMP assumes phylogenetic conservatism of niche (significant phylogenetic signal) within short phylogenetic distances, in other words, members in the same bin have “niche differences” significantly correlated with their phylogenetic distances. As we replied to the question B33, if strong competition predominates, this assumption does have risk to be disrupted. But the competition under natural settings is mostly likely not so strong to disrupt the assumption as theoretically predicted, otherwise phylogenetic signal within short phylogenetic distances would not be so widely observed (e.g. Wang et al 2013; Stegen et al 2012; Tripathi et al 2018).

References:

- [1] Wang, J. J. et al. Phylogenetic beta diversity in bacterial assemblages across ecosystems: deterministic versus stochastic processes. *ISME J* 7, 1310-1321, doi:10.1038/ismej.2013.30 (2013).
- [2] Stegen, J. C., Lin, X., Konopka, A. E. & Fredrickson, J. K. Stochastic and deterministic assembly processes in subsurface microbial communities. *ISME J* 6, 1653–1664 (2012).
- [3] Tripathi, B. M. et al. Soil pH mediates the balance between stochastic and deterministic assembly of bacteria. *ISME J* 12, 1072-1083, doi:10.1038/s41396-018-0082-4 (2018).

(B36) Lines 310-319: Given the majority (too much in my opinion) of your results are focused how iCAMP outperforms QPEN, I feel you must do a better job justifying why these comparisons are so important. Why the disdain for QPEN? It you feel that QPEN is just bad at estimating these factors and you want to prevent folks from using so that they do not have spurious results and them to infer inappropriate patters, that is one thing. But it just reads like you have QPEN for an unknown reason. I think it is important for the reader to know why. This is imperative to put your advancements in the proper light.

Reply: iCAMP is developed based on the defect of QPEN (we clearly stated this in the Introduction), and borrows a lot from the null model framework in QPEN. We believe when a new method is developed based on some defect of a previous method, it is compelling to compare its performance with the previous method. When NST was developed, we did compare it with the previous ST (Ning et al 2019). When O2PLS was developed, it was compared with PLS (Trygg et al 2003). When an ASV-based approach (e.g. DADA2) was developed, it was compared with the OTU-based approach (Challahan et al 2017). When VSEARCH is developed, it was compared with USEARCH (Rognes et al 2016).

Developers surely have intention to replace the previous method with a new method, at least under some situations, considering the defect(s) they found in the previous method. Otherwise, it makes no sense to develop a new method. Although “disdain” and “prevent” are relatively strong words, we do intend to replace QPEN with iCAMP to estimate relative importance of different assembly processes. It does not mean to entirely discard QPEN, and surely not to disdain it.

Actually, QPEN is a fundamental part of iCAMP. Our comparison is just like comparing O2PLS with PLS or comparing NST with ST, which did not mean to disdain PLS or ST, but to further develop them. The intention is positive rather than negative.

By considering Reviewer's concerns, this paragraph was deleted from Discussion to less emphasize their differences in performance. The deletion of this paragraph does not affect the scientific integration and presentation of this study.

References:

- [1] Ning, D., Deng, Y., Tiedje, J. M. & Zhou, J. A general framework for quantitatively assessing ecological stochasticity. *Proceedings of the National Academy of Sciences of the United States of America* 116, 16892-16898, doi:10.1073/pnas.1904623116 (2019).
- [2] Trygg J, Wold S. O2-PLS, a two-block (X–Y) latent variable regression (LVR) method with an integral OSC filter. *Journal of Chemometrics* 17:53-64, doi:10.1002/cem.775 (2003)
- [3] Callahan, B., McMurdie, P. & Holmes, S. Exact sequence variants should replace operational taxonomic units in marker-gene data analysis. *ISME J* 11, 2639–2643, doi:10.1038/ismej.2017.119 (2017).
- [4] Rognes T, Flouri T, Nichols B, Quince C, Mahé F. VSEARCH: a versatile open source tool for metagenomics. *PeerJ*. 4:e2584, doi:10.7717/peerj.2584 (2016)

(B37) Lines 319-323: It is a bit anthropomorphizing to claim that these grasslands are not stressful. If there were not stressors, there would be no competition which of course does not happen.

Reply: This paragraph was deleted.

(B38) Lines 339-344: This may be a bit speculative based on one grassland study.

Reply: We agree. Since the potential implication is not main point in this study, we decide to delete these sentences.

(B39) Methods:

Lines 362-363: I am not sure what I think of the 'core taxa' label – core taxa in microbial ecology generally means one that is ubiquitous across the experimental framework. I worry using this same terminology to mean something else is unnecessarily confusing.

Reply: We agree. It is not proper to use “core taxa”. We revised it to “centroid taxa”.

(B40) Lines 366-367 and Lines 376-377: This gets at an earlier point I made, if bins are merged, don't you lose the power to assume niche conservation? It might be better to omit these bins from analysis than combine them.

Reply: We agree. Omitting these bins could be the best choice in some cases. However, because the sequencing depth in many studies are not high level. Omitting such data could result in very

few bins left. For instance, in the simulated data, omitting those bins will discard a large portion of species (up to 100% in some simulated situations). This can also happen in real data if the diversity is at the similar level as the simulated data. Thus, we developed the algorithm to merge small bins to their nearest relatives. Please find more details about this issue in our reply to the question B2.

Given the complex world of microbial communities, we do realize omitting the bins can be better in some cases, for example, the omitted bins have only 1% taxa with <0.001% relative abundance and not relevant. Thus, we added an option to omit the small bins (“omit.option” in function “icamp.big” in iCAMP package), and if a user uses this option, it will firstly output the information of omitted bins to facilitate the user to decide. We added a sentence in method part, Line 398-399 [tracking version Line 484-486]“Although not used in this study, another option is also provided in our tool to omit small bins when they are negligible.”

(B41) Lines 381-394: I would like to learn more about this ‘niche preference’ approach. It does not seem to be that you would have enough information to identify a niche preference, and what does this even mean? This is a bit nebulous but it appears crucial to your approach so it must be treated as such and this require justification for these assumptions.

Reply: We agree. In the revision, we added a tool to the package to calculate niche difference (the function “dniche”) and test within-bin phylogenetic signal (the function “ps.bin”), which will work if key environmental factors are measured. If the environmental data are not available or may not include the key factor(s), we suggested another way based on stochasticity estimation. We also mentioned the assumption and the alternative way in the method part (as below) and the Supplementary text Line 35 to 41.

Main text Line 416-420 [tracking version Line 503-507]: The index “niche value” works under the assumption that relative abundances can represent the fitness of a taxon and the available environmental factors can measure the niche profile. Otherwise, an alternative way should be considered, i.e. choose the n_{min} value that makes the estimated relative importance of stochastic processes close to the stochasticity assessed by referable approaches (e.g. pNST).

(B42) Lines 431-434: I think this would be improved if you also did a iterative approach on the grassland to explore how many randomization steps are suggested for real data.

Reply: We agree, but not able to do so many times for the real data (too time consuming). We repeated iCAMP and QPEN for 3 times, with the commonly used 1000-time randomization in null models. iCAMP can estimate relative importance of each process in the 780 pairwise comparisons among the 40 samples. In the 780 results of each process, the standard deviation among the 3 times ranges from 0 to 6.8% (which is 8.8% or 3.9% if randomizing 200 or 3000 times, respectively), with IQR values less than 1.02% (which is 1.5% or 0.60% if randomizing 200 or 3000 times) different processes, indicating sufficient reproducibility. QPEN can identify one dominating process for each of the 780 pairwise comparisons. In the 780 estimates, only 1.9% (which is 4.7% or 1.4% if randomizing 200 or 3000 times) estimates are different among

the 3 times. These results were added to supplementary text Line 88-97 [tracking version supp. Line 95-104] in the revision. The 1000-time randomization is acceptable.

(B43) Lines 542-543: Is it reasonable to assume no dispersal limitations? Especially since the authors claim that iCAMP is suitable for animal-based systems as well?

Reply: Here we mean “at moderate dispersal rate rather than limited or intensive dispersal”(main text Line 591-592 [tracking version Line 679-680]). This part is about simulation model. Simulation is always simplified or idealized from the real world. We have to simulate species “solely” controlled by a certain process without other processes, then combine species controlled by different processes together in different ratios. This is the only way we can exactly define the relative importance of a certain process in the simulation model, although the “solely controlled” is definitely not reasonable in the real world.

In real world, we believe a large number (if not all) of microorganisms should have some dispersal limitations, particularly in soil. But microbial dispersal can be moderate or higher without obvious limitation in some ecosystems, e.g. free-living bacteria in running rivers, or bioreactors. This could also happen in animal ecology. When we consider birds in a region or rats in a plain, is it acceptable to say they are not under obvious dispersal limitations?

By the way, a possible misunderstanding is to take the low values for dispersal limitation in iCAMP results as little or no dispersal limitation. The low values should actually be interpreted as low influence of dispersal limitations than other processes, i.e. there can be a lot of limitations, but they are not the major reasons of community structure variations.

(B44) Lines 618-619: This sentence is odd and needs to be rewritten for clarity.

Reply: We revised it to “Then the V4 region of 16S rRNA gene was amplified from soil community DNA and sequenced with Illumina MiSeq System, and the outputs were processed as described previously”(main text Line 672-674 [tracking version Line 762-765]).

Reviewers' Comments:

Reviewer #1:

Remarks to the Author:

I am overall happy with the revision. I think the authors did a good job answering our remarks. I do still have a methodological concern and very minor comments. I was unable to access the code of the analysis.

Regarding my remark about examining between treatments beta-diversities, I am not quite convinced that neglecting them is opportun... But fine. I would recommend that the authors make explicit in the discussion and or methods that they make inference about the within-treatment and within-year heterogeneity of communities. When reading the current text, I feel that the reader may still think that it is the community turnover between treatment/years that is examined.

Regarding the use of effect size threshold, have you evaluated the shape of the null distributions for beta NRI and RC ? As you state, the threshold for betaNRI assumes that the null model distributions are normally distributed, (a threshold of $|1.96|$ corresponds to a double tailed test with a 5% error rate). It is widely used in diversity pattern analyses, as the authors know, but it relies on the assumption, rarely checked, that null distributions are akin to normal distributions.

I would recommend that the authors check where the observed beta-diversity values fall in their respective null distribution (% of null values above the observed diversity value) to determine "significance" instead of assuming that the null distribution is normal and that the effect size values give that information. In other words, the null distribution of your indices is likely skewed and the use of effect sizes will produce false positives and/or negatives.

For more details, please read the paper of Veech 2012 'Significance testing in ecological null models' – Theoretical Ecology. To quote him: "For example, the net relatedness index and nearest taxon index of Kembel and Hubbell (2006) often produce distributions that are right skewed (Swenson et al. 2006)."

Examining all this may improve the accuracy of iCAMP, as according to your results, your method has trouble detecting homogeneizing selection and (to a lesser extent) high dispersal (fig S12) for low phylogenetic signal.

I. 50. Trade-offs are not biotic interactions.

I. 52 species dynamics "are" largely (...)

I. 53 : "After intensive decade's debates" – there is something wrong with this expression.

I.155: I would use put the raw stat value of both approaches in parenthesis. It is confusing to read precision (120.2%) instead of precision (X% against Y%)

I. 252 double parentheses and double comma to correct.

Loïc Chalmandrier

Reviewer #2:

Remarks to the Author:

The authors provide a much-improved MS on a novel way to account for different assembly processes in microbial communities. While the authors' addressed my concerns in the first MS, some of the revisions are not as detailed as one might have hoped. The addition of more simulations to strengthen the reliance on baseline assumptions was much needed and I am pleased with this. Overall, this is a strong and potentially important contribution. There are still a few reservations I have (see below), namely, I am not convinced that binning OTUs into phylogenetic bins is a reasonable way to approach this question as cross-bin interactions likely play a large role in assembly dynamics, and this is unaccounted for. Phylogenetic binning certainly would decrease computational time and allow for better multi-threading, but I am not yet convinced that this is a reasonable approach, and I was hoping for more discussion in this revision. One additional thought is how would iCAMP respond to fungal ITS data? These can be binned phylogenetically but reasonable trees can not be generated, so iCAMP would not perform well (I would imagine). I think it would be worth mentioning this required alignable datasets to prevent potential inappropriate use other researchers.

Line 31: the phrase "which is largely upon the Bacillales" makes little sense.

Line 45: Should citation be numeric?

Line 58: Is niche synonymous with deterministic here? These can be two different things.

Line 79: "finer biological levels" is an odd phrase with no real operational definition.

Is binning based on phylogenetic similarity a realistic way to infer assembly – that is, are phylogenetically similar lineages going to necessarily respond differently than other bins? I find this unlikely.

Line 150: multiple commas

Lines 170-175: I real systems, consortia of lineages that respond similarly are likely to be of low phylogenetic relatedness (you likely have co-coculating or associated taxa from many phyla for instance). Thus, by the author's own statement, iCAMP only preforms so-so in more theoretically realistic scenarios (still better than other analytical tools, however). Given this, I would have appreciated a discussion on how this is an incremental improvement in some circumstances. But they claim that this is a "substantial" improvement. It is no doubt an improvement, but I am not sure how "substantial" it is. One must be careful not to overextend the level of inference the data show.

Responses to Reviewers' comments

Manuscript number: NCOMMS-20-03950A

A. Reviewer #1 (Remarks to the Author):

(A1) I am overall happy with the revision. I think the authors did a good job answering our remarks. I do still have a methodological concern and very minor comments. I was unable to access the code of the analysis.

Reply: We are grateful for all the comments and suggestions. To ensure accessibility, we have made iCAMP as a R package and ready to submit to the Comprehensive R Archive Network (CRAN). It will be submitted once the CRAN submission website back to normal (Aug 25). Meanwhile, iCAMP is also available in our web-based pipeline (<http://ieg3.rccc.ou.edu:8080>), which facilitates general users who are not familiar with programming languages. All custom scripts are available from GitHub now. We also plan to publish the all-in-one zip file as Supplementary Code, which contains the current version of iCAMP, detailed ReadMe file, and an example of codes, input, and output files (see ‘Code Availability’).

(A2) Regarding my remark about examining between treatments beta-diversities, I am not quite convinced that neglecting them is opportun... But fine. I would recommend that the authors make explicit in the discussion and or methods that they make inference about the within-treatment and within-year heterogeneity of communities. When reading the current text, I feel that the reader may still think that it is the community turnover between treatment/years that is examined.

Reply: Revised. We added “with focus on within-treatment spatial turnovers” in the Result (Line 193 [tracking version Line 206]), and emphasized “the empirical study only investigated within-treatment spatial turnovers at each time point” in the Method part (Line 720-721 [tracking version Line 745-746])

(A3) Regarding the use of effect size threshold, have you evaluated the shape of the null distributions for beta NRI and RC ? As you state, the threshold for betaNRI assumes that the null model distributions are normally distributed, (a threshold of |1.96| corresponds to a double tailed test with a 5% error rate). It is widely used in diversity pattern analyses, as the authors know, but it relies on the assumption, rarely checked, that null distributions are akin to normal distributions.

I would recommend that the authors check where the observed beta-diversity values fall in their respective null distribution (% of null values above the observed diversity value) to determine “significance” instead of assuming that the null distribution is normal and that the effect size values give that information. In other words, the null distribution of your indices is likely skewed and the use of effect sizes will produce false positives and/or negatives.

For more details, please read the paper of Veech 2012 ‘Significance testing in ecological null models’ – Theoretical Ecology. To quote him: “For example, the net relatedness index and nearest taxon index of Kembel and Hubbell (2006) often produce distributions that are right skewed (Swenson et al. 2006).”

Examining all this may improve the accuracy of iCAMP, as according to your results, your method has trouble detecting homogenizing selection and (to a lesser extent) high dispersal (fig S12) for low phylogenetic signal.

Reply: We agree this is an important point. β NRI is a standard effect size between observed and null values of β MPD. If the distribution of null β MPD is skew, the β NRI threshold 1.96 may misestimate the significance of observed β MPD. The modified RC is not standard effect size, but calculated from the direct count of null values higher, lower, and equal to the observed value (Eq. 7 and 8). Thus, RC does not require any assumption on null distribution, but it does split the cases when null values are equal to the observed value, which may lead to some misestimation of significance but very slight.

As the reviewer recommended, we tested the option to calculate significance by directly counting the percentage of null values higher or lower than the observed beta-diversity value, i.e., non-parametric one-tail confidence level, so-called Confidence (Eq. S1, S2). Under normal distribution, the standard effect size threshold 1.96 is equivalent to the one-tail Confidence value 0.975. We changed iCAMP to use the Confidence > 0.975 for significance test in both phylogenetic and taxonomic null model analysis. Then we compared iCAMP results and performance with those from original iCAMP using β NRI+RC. Interestingly, iCAMP performance is almost the same. The performance indexes showed concordance and Pearson's correlation coefficient > 0.99 and negligible difference (Cohen's $d < 0.11$, Wilcoxon $P > 0.93$) between using β NRI+RC and Confidence, at community and bin levels. The estimations of process importance are also largely consistent between using β NRI+RC and Confidence, in both simulated and empirical data. Considering their performance, popularity, and links to the previous approach (QPEN), we still mainly used β NRI and RC for iCAMP analysis in this study.

Nevertheless, we agree with the reviewer that β NRI can have the risk of misestimation, given the fact that microbial communities are so diverse and complicated. Thus, we emphasized this problem in the main text (Line 119-123 [tracking version Line 129-133]) and added a detailed description in the Supplementary Note 1 'Null model significance testing indexes' section. In iCAMP package, we added "Confidence" as the default option to identify significant differences between null and observed values. We also added a function "null.norm" to test the normality of null distributions and a function "change.sigindex" to quickly change the significance-assessing index.

(A4) l. 50. *Trade-offs are not biotic interactions.*

Reply: Deleted.

(A5) l. 52 *species dynamics "are" largely (...)*

Reply: Revised.

(A6) l. 53 : *"After intensive decade's debates" – there is something wrong with this expression.*

Reply: Revised to "After intensive debates in 2000s, ...".

(A7) l.155: *I would use put the raw stat value of both approaches in parenthesis. It is confusing to read precision (120.2%) instead of precision (X% against Y%)*

Reply: Revised.

(A8) l. 252 double parentheses and double comma to correct.

Reply: Revised and checked through all text.

B. Reviewer #2 (Remarks to the Author):

(B1) The authors provide a much-improved MS on a novel way to account for different assembly processes in microbial communities. While the authors' addressed my concerns in the first MS, some of the revisions are not as detailed as one might have hoped. The addition of more simulations to strengthen the reliance on baseline assumptions was much needed and I am pleased with this. Overall, this is a strong and potentially important contribution. There are still a few reservations I have (see below), namely, I am not convinced that binning OTUs into phylogenetic bins is a reasonable way to approach this question as cross-bin interactions likely play a large role in assembly dynamics, and this is unaccounted for. Phylogenetic binning certainly would decrease computational time and allow for better multi-threading, but I am not yet convinced that this is a reasonable approach, and I was hoping for more discussion in this revision.

Reply: We agree cross-bin interactions are important and need to discuss. We emphasized it in the end of our discussion and added detailed discussion in Supplementary Note 3.

Main text Line 376-382 [tracking version Line 391-398]:

In addition, although built on within-bin beta diversity, iCAMP should generally be able to capture important cross-bin selection (Supplementary Note 3). However, iCAMP might underestimate selection when cross-bin selection does not lead to detectable within-bin difference. Thus, further developments are needed ... and by integrating functional traits (genes) and network approaches with iCAMP to ... capture special cross-bin selection.

(B2) One additional thought is how would iCAMP respond to fungal ITS data? These can be binned phylogenetically but reasonable trees can not be generated, so iCAMP would not perform well (I would imagine). I think it would be worth mentioning this required alignable datasets to prevent potential inappropriate use other researchers.

Reply: We agree this point should be emphasized. We added a sentence in the Method part as below. Although still possible, building fungal phylogenetic trees from ITS sequences does require particular approaches (e.g. Fouquier et al 2016 Microbiome; Nuccio et al 2016 Ecology).

Main text Line 422-423 [tracking version Line 436-440]:

All binning algorithms demand a reliable phylogenetic tree, which may not be applicable or need particular approaches for highly divergent marker genes (e.g. ITS).

References:

- [1] Fouquier, J. et al. ghost-tree: creating hybrid-gene phylogenetic trees for diversity analyses. *Microbiome* 4, 11, doi:10.1186/s40168-016-0153-6 (2016).
- [2] Nuccio, E. E. et al. Climate and edaphic controllers influence rhizosphere community assembly for a wild annual grass. *Ecology* 97, 1307-1318, doi:10.1890/15-0882.1 (2016).

(B3) Line 31: the phrase “which is largely upon the Bacillales” makes little sense.

Reply: Revised to “The selection was primarily imposed on Bacillales.”

(B4) Line 45: Should citation be numeric?

Reply: Revised and checked the same issue through all text.

(B5) Line 58: Is niche synonymous with deterministic here? These can be two different things.

Reply: In this sentence, “niche” represents deterministic processes, including environmental filtering and various biological interactions, as mentioned in the previous paragraph. The sentence is about to introduce Vellend’s framework which integrated processes in niche and neutral theories. Thus, “To unify niche and neutral perspectives” should be appropriate in the context here.

(B6) Line 79: “finer biological levels” is an odd phrase with no real operational definition.

Reply: Revised to “finer biological organization levels ...”, based on the concept of biological organization level from Evans (1951, 1956).

References:

- [1] Evans, F. C. Ecology and urban areal research. *Scientific Monthly*, 73 (1951).
- [2] Evans, F. C. Ecosystem as basic unit in ecology. *Science*, 123, 1127–8, doi:10.1126/science.123.3208.1127 (1956)

(B7) Is binning based on phylogenetic similarity a realistic way to infer assembly – that is, are phylogenetically similar lineages going to necessarily respond differently than other bins? I find this unlikely.

Reply: We believe binning based on phylogenetic similarity is a realistic and reasonable way to infer assembly mechanisms. We explained the rationale in the Introduction (Line 80-90 [tracking version Line 89-100]) and Discussion part (Line 334-345 [tracking version Line 348-359]) in detail. The point is supported by the promising performance of iCAMP on our simulated and empirical data. Also, binning does not necessarily require “phylogenetically similar lineages are going to respond differently than other bins”. On the contrary, in this study, iCAMP identified the same or similar response of many different bins (e.g. the 59 bins dominated by homogeneous selection). In addition, a bin is allowed to respond differently under different conditions, e.g. iCAMP identified two bins that switched from drift-controlled in control plots to selection-controlled under warming.

(B8) Line 150: multiple commas

Reply: Revised and checked the same issue through all text.

(B9) Lines 170-175: In real systems, consortia of lineages that respond similarly are likely to be of low phylogenetic relatedness (you likely have co-occurring or associated taxa from many phyla for instance). Thus, by the author's own statement, iCAMP only performs so-so in more theoretically realistic scenarios (still better than other analytical tools, however). Given this, I would have appreciated a discussion on how this is an incremental improvement in some circumstances. But they claim that this is a "substantial" improvement. It is no doubt an improvement, but I am not sure how "substantial" it is. One must be careful not to overextend the level of inference the data show.

Reply: Sorry for the confusion. In this paragraph, "substantial" is based on the comparison of "quantitative" performance, i.e. quantitative accuracy and precision, which range from 0.71 to 1.00 for iCAMP, but down to negative values for QPEN. Collectively, quantitative accuracy and precision of iCAMP for each process are 129% higher than QPEN, on average. We believe this can be concluded as a "substantial" improvement in quantitative estimation. We revised as follows to avoid confusion.

Main text Line 186-189 [tracking version Line 197-201]:

Nevertheless, the quantitative performance of iCAMP remained relatively high for all processes under all scenarios, with quantitative accuracy and precision 0.71 - 1.00 (averagely 129% higher than QPEN), indicating that iCAMP can substantially improve the quantitative estimation of community assembly processes.

In the whole study, iCAMP showed high overall performance scores which are 10-160% higher than the previous method, provided detailed mechanisms down to each phylogenetic bin which is not possible by previous approaches, and obtained more reasonable estimation in empirical data. Collectively, we insist iCAMP provides substantial improvement.

The "so-so" performance of iCAMP only appears in "qualitative" sensitivity and precision under low phylogenetic signal scenario, which is not due to low phylogenetic relatedness but low niche conservatism under the scenario. This may not be a big problem for iCAMP, because in empirical studies, we only use iCAMP quantitative results, not "qualitative" results.

In the empirical data (a real system), we do see the co-occurrence of different phyla. However, the importance of homogeneous selection (~38%) is much higher than heterogeneous selection (<0.4%, Fig. 3a, b). This result indicates that within the phylogenetic signal threshold, phylogenetic relatedness is mostly higher than the null expectation.